*Report*

# PV-1: a novel molecular prognostic marker of distant metastases in various solid tumors

Chiara Pozzi [1,3], Riccardo Sarti [1,2,3], Antonino Lo Cascio[1], Stefanos Bonovas [1,2], Luca Tiraboschi [1,2], Michela Lizier [1], Alice Bertocchi[1], Rosalba Torrisi[1], Bethania Fernandes [1], Piergiuseppe Colombo [1,2], Pietro Diana[1,2], Emilia Maria Cristina Mazza[1], Marina Valeri[1,2], Salvatore Lorenzo Renne [1,2], Ferdinando Carlo Maria Cananzi [1,2], Laura Samà[1], Alexia Bertuzzi[1], Corrado Tinterri[1,2], Massimo Lazzeri [1] & Maria Rescigno [1,2]✉

## Abstract

**Identification of biomarkers for the hematogenous spreading of cancer cells is of paramount prognostic and therapeutic value. We showed that Plasmalemma Vesicle Associated Protein-1 (PV-1) serves as a marker of increased blood vessel permeability and is an independent predictor of colorectal cancer dissemination. This study investigates whether PV-1 can also act as a prognostic marker for distant metastases in other solid tumors. We analyzed samples from 134 patients: 30 luminal breast cancer (BC), 52 clear cell renal cell carcinoma (ccRCC), and obtained preliminary data from 52 soft tissue sarcomas (STS). A higher frequency of PV-1+ endothelial cells was significantly associated with metastatic progression in luminal BC and ccRCC. Moreover, the frequency of PV-1+ cells emerged as a significant prognostic factor for metastasis-free survival in both luminal BC and ccRCC. Further research is needed to validate PV-1's prognostic utility, as including it at diagnosis may change the management of these patients and should allow stratification for more aggressive therapies or for closer follow-ups to promptly intervene in case of metastases development.**

**Keywords** PV-1 (PLVAP); Metastasis-free Survival; Endothelial Permeability; Solid Tumors; Biomarker
**Subject Categories** Biomarkers; Cancer; Vascular Biology & Angiogenesis

## Introduction

We have recently demonstrated a positive association between endothelial disruption and hematogenous dissemination in colorectal cancer (CRC) patients (Bertocchi et al, 2021). Notably, we previously identified a Gut Vascular Barrier (GVB) that functions alongside the intestinal epithelial barrier, serving as a regulatory barrier to prevent the translocation of bacteria and large food proteins into the bloodstream. Under hazardous conditions, impairment of the GVB leads to altered blood vessel permeability, as evidenced by increased Plasmalemma Vesicle Associated Protein-1 (PV-1) detection, a type II transmembrane glycoprotein encoded by the *PLVAP* gene (Spadoni et al, 2015). PV-1 is a known molecular constituent of fenestral and stomatal diaphragms that cover the pores of the discontinuous endothelium (Stan, 2007; Stan et al, 2004). Interestingly, CRC patients exhibiting elevated PV-1 detection in the primary tumor, independently on other factors, have twice the likelihood of developing distant metastasis in the follow-up time. Thus, PV-1 acts as a marker of blood endothelial permeability impairment and emerges as a prognostic marker for distant metastases in CRC (Bertocchi et al, 2021). Building on this evidence, we evaluated whether PV-1 detection could serve as a marker for distant metastases also in other solid tumors known to metastasize via the hematogenous route, such as luminal breast cancer and clear cell renal cell carcinoma (ccRCC).

Breast cancer is still the leading cause of cancer-related deaths for women worldwide (Loibl et al, 2024). Notably, approximately two-thirds of breast cancers exhibit hormone receptor positivity (HR+) for estrogen and progesterone (Howlader et al, 2014). HR+HER2- tumors, categorized as Luminal A and B, generally display a lower relapse rate within 3–4 years after diagnosis as compared to triple-negative and HER2 enriched breast cancer (Cossetti et al, 2015). Endocrine therapy (ET) is the mainstay of adjuvant therapy for these patients. However, a subset of these patients proves resistant to ET and recurs within a span of 2 years and is classically defined as primary endocrine resistant (Cardoso et al, 2017). Tumor extension, in particular the involvement of axillary lymph nodes is still considered as a pivotal and enduring prognostic factor for recurrence, despite about 30% of patients without lymph node involvement at diagnosis may relapse (Cardoso et al, 2017). In addition to tumor extension a relevant prognostic role has been attributed to biologic factors such as proliferation rate and genomic tumor characteristics (intrinsic tumor subtype, genomic signatures, etc.), which may also drive the decision to candidate patients to adjuvant chemotherapy in

[1]IRCCS Humanitas Research Hospital, via Manzoni 56, 20089 Rozzano, Milan, Italy. [2]Department of Biomedical Sciences, Humanitas University, Via Rita Levi Montalcini 4, 20072 Pieve Emanuele, Milan, Italy. [3]These authors contributed equally: Chiara Pozzi, Riccardo Sarti. ✉E-mail: maria.rescigno@hunimed.eu

addition to ET (Loibl et al, 2024). Still, many patients who are initially considered to have a favorable outcome unexpectedly relapse, highlighting the urgent need to identify additional prognostic factors that may help predict early relapse in this tumor subtype.

Renal Cell Carcinoma (RCC) accounts for roughly 3% of all cancers, and it stands as the predominant parenchymal lesion in the kidney, comprising about 90% of all kidney malignancies. There are three main RCC types, and among these, clear cell Renal Cell Carcinoma (ccRCC) is the most prevalent and aggressive form of renal malignancy (70–80%) with up to 30% of ccRCC patients developing metastases (Ljungberg et al, 2022; Lotan and Margulis, 2019). Despite advancements in early ccRCC detection and primary tumor removal through surgery, existing stratification models struggle to accurately predict metastatic occurrences. Consequently, active surveillance remains the primary management approach (Capitanio and Montorsi, 2016; Rini et al, 2009). This underscores the urgent need for novel molecular markers to enhance patient risk stratification (Williamson et al, 2019). The transition of renal cell carcinoma to a metastatic state occurs when malignant cells disseminate from the primary tumor to distant organs, primarily via the systemic blood circulation (Lotan and Margulis, 2019), which is why we hypothesize that PV-1 could serve as a marker for distant metastases also in this type of tumor.

Another tumor that metastasizes through the hematogenous route is the soft tissue sarcoma (STS) (Pennacchioli et al, 2012). Soft tissue sarcomas (STS) represent a rare and heterogeneous group of diseases, constituting less than 1% of all adult cancers. This category encompasses over 70 histologic subtypes, each exhibiting distinct biological behaviors and responses to chemotherapy. In the last decades, with few exceptions for some histotypes, there have been no real breakthroughs, neither in the understanding of sarcomagenesis nor in the treatment of the different types of STS. Complete surgical resection remains the unique chance to effectively cure sarcoma, but approximately one-third will eventually develop metachronous distant metastases (Posch et al, 2017). Unfortunately, patients with metastatic STS face limited systemic treatment options and generally experience a poor prognosis, with a median survival of about 1 year (Italiano et al, 2011; Van Glabbeke et al, 1999). Current conventional prognostic factors cannot properly depict sarcoma-complexity thus are not sufficient for accurately predicting the incidence of metastasis (Weaver et al, 2020). Hence, there is an urgent need for new molecular markers to enhance patient classification. 52 soft tissue sarcoma (STS) samples were analyzed in this study, and while the results are promising and offer interesting insights, they remain preliminary due to the heterogeneity and small size of the cohort and are therefore presented in the Appendix.

Hence, in this study, we aim to elucidate the value of endothelial cell-associated PV-1 as a biomarker of distant metastases in luminal breast cancer, ccRCC, and, albeit preliminarily, in STS as well.

## Results and discussion

To assess the occurrence of PV-1+ cells in metastatic and non-metastatic patients with various tumor types, we conducted a retrospective analysis using formalin-fixed paraffin-embedded (FFPE) resected cancer samples. Our study encompassed a total of 134 samples, comprising 30 cases from luminal breast cancer patients (18 metastatic and 12 non-metastatic), 52 from clear cell renal cell carcinoma (ccRCC) patients (22 metastatic and 30 non-metastatic), and 52 sarcoma patients, including 28 metastatic and 24 non-metastatic cases.

### Luminal breast cancer

We analyzed tumor tissue obtained at diagnosis in women with luminal breast carcinoma who exhibited primary resistance to endocrine therapy (defined as relapse within 2 years of initiating treatment) (metastatic, $n = 18$) and compared it with tumor tissue from women with similar tumor stage and treatment (endocrine therapy plus chemotherapy when indicated) who remained disease-free after 5 years (non-metastatic, $n = 12$) (Table 1).

A higher frequency of PV-1+ cells among CD-31+ endothelial cells was observed in the primary tumor of metastatic luminal breast cancer patients compared to non-metastatic patients ($P = 0.006$, Fig. 1A,B). Univariable Cox proportional hazards (PH) regression analyses were performed for each clinical variable. The variables included: percentage of PV-1+ cells (analyzed per 10-unit increase), number of positive lymph nodes, tumor size (per 1-cm increase), histological features—namely estrogen receptor (ER), progesterone receptor (PgR), and Ki-67 expression (per 10-unit increase), tumor stage (a variable that encompasses both tumor size and the number of positive lymph nodes), patient age (per 10-year increase), and number of comorbidities (Table 2). Among the 30 luminal breast cancer patients, 16 had no comorbidities, while 11 reported a single comorbidity (dyslipidemia $n = 4$; arterial hypertension $n = 3$; bronchial asthma $n = 1$; vascular disease $n = 1$; hyperthyroidism $n = 2$), and 3 patients had multiple comorbidities. A multivariable Cox PH model with backward stepwise elimination identified the percentage of PV-1+ cells as the sole significant independent prognostic factor for metastasis-free survival (HR 1.51, 95% CI 1.02–2.22, $P = 0.038$) (Table 2). The PH assumption and other model assumptions were adequately met. In addition, a time-dependent ROC analysis at 5 years was performed for each of the above clinical variables. The highest AUC values were observed for tumor size (AUC = 0.713), percentage of PV-1+ cells (AUC = 0.694), and Ki-67 (AUC = 0.630). Caution is warranted in interpreting ER/PgR AUCs due to limited sample size (Table 3; Fig EV1).

We further explored PV-1 as a prognostic marker by leveraging publicly available bioinformatic tools and datasets. Specifically, we conducted Kaplan–Meier survival analyses using the KM Plotter tool (Győrffy, 2021) to assess PV-1 expression levels (at the gene level, *PLVAP*, or protein level, PV-1) in luminal breast cancer patients. When focusing on mRNA gene chip data from Luminal A and Luminal B breast cancer patients treated with endocrine therapy, the results aligned with our findings, showing consistent trends for distant metastasis-free survival (DMFS) and relapse-free survival (RFS). Notably, patients with high expression of the *PLVAP* gene exhibited shorter DMFS and RFS compared to those with low *PLVAP* expression (Fig. EV2A). In addition, we examined protein expression data of PV-1 in breast cancer using the Tang et al, 2018 dataset (Tang et al, 2018), which includes 65 tumor samples. While the overall survival (OS) plot showed a trend consistent with our analysis, the $P$ value was not significant. It is worth noting that this analysis could not be restricted to luminal

**Table 1.** Demographics and clinical characteristics of the luminal breast cancer patients (n = 30).

| | | Overall | Non-metastatic | Metastatic | P value | Test |
|---|---|---|---|---|---|---|
| n | | 30 | 12 | 18 | | |
| Age, mean (SD) | | 56.1 (11.7) | 54.7 (8.7) | 57.1 (13.5) | 0.551 | t test |
| Percentage of PV-1+ cells, median [Q1, Q3] | | 90.0 [72.5, 97.5] | 70.0 [60.0, 90.0] | 90.0 [90.0, 100.0] | **0.006** | Mann–Whitney |
| Follow-up time or metastasis onset[a] (months), median [Q1, Q3] | | 26.1 [21.4, 60.0] | 60.0 [60.0, 60.0] | 24.0 [17.1, 25.6] | **<0.001** | Mann–Whitney |
| Tumor stage, n (%) | I | 4 (13.3) | 2 (16.7) | 2 (11.1) | 0.842 | Chi-squared |
| | II | 17 (56.7) | 7 (58.3) | 10 (55.6) | | |
| | II | 9 (30.0) | 3 (25.0) | 6 (33.3) | | |
| Tumor dimension (mm), median [Q1, Q3] | | 23.0 [16.0, 35.0] | 19.0 [13.0, 24.0] | 27.0 [21.0, 40.0] | 0.057 | Mann–Whitney |
| Percentage of ER, median [Q1, Q3] | | 92.5 [90.0, 95.0] | 92.5 [90.0, 95.0] | 92.5 [90.0, 95.0] | 0.851 | Mann–Whitney |
| Percentage of PgR, median [Q1, Q3] | | 50.0 [5.0, 85.0] | 67.5 [4.0, 95.0] | 50.0 [30.0, 70.0] | 0.563 | Mann–Whitney |
| Percentage of Ki-67, median [Q1, Q3] | | 20.0 [11.0, 35.0] | 12.5 [10.0, 30.0] | 30.0 [15.0, 35.0] | 0.081 | Mann–Whitney |
| Menopause, n (%) | | 21 (70.0) | 8 (66.7) | 13 (72.2) | 1.000 | Fisher's exact |
| Tumor location, n (%) | Left breast | 14 (46.7) | 6 (50.0) | 8 (44.4) | 1.000 | Chi-squared |
| | Right breast | 16 (53.3) | 6 (50.0) | 10 (55.6) | | |

[a]Follow-up time refers to non-metastatic patients, while metastasis onset refers to metastatic patients. There were 3 missing Ki-67 values and 1 missing PgR value, which were imputed with the mean value of the other patients.
Bold values indicate statistical significance $P \leq 0.05$.

subtypes alone, as the dataset also includes HER2-positive and triple-negative breast cancer (TNBC) subtypes (Fig. EV2B). To further evaluate the prognostic value of PV-1 expression at the gene level, we performed a ROC curve analysis using the ROC Plotter tool. This tool uses 5-year relapse-free survival (RFS) as the clinical endpoint and constructs ROC curves by classifying patients based on their clinical outcome—those without relapse within 5 years and those with relapse within 5 years. *PLVAP* gene expression was assessed as a continuous predictor of outcome. The analysis revealed limited prognostic accuracy for *PLVAP* in luminal A patients (AUC = 0.533) and moderate accuracy in luminal B patients (AUC = 0.608), both treated with endocrine therapy (Fig. EV3).

The significant correlation between high PV-1+ cell frequency and metastatic progression supports the hypothesis that endothelial disruption facilitates the hematogenous dissemination. PV-1 emerges as an independent prognostic factor of metastasis-free survival in luminal breast cancer, underscoring its potential utility as a prognostic marker in this cancer subtype.

## Clear cell renal cell carcinoma

A total of 52 tumor samples from patients undergoing upfront open or robotic partial or radical nephrectomy for clear cell renal cell carcinoma (ccRCC) were analyzed, subdivided into 22 metastatic and 30 non-metastatic cases. Non-metastatic patients did not develop metastases within a median follow-up period of 15 years. As shown in Table 4, univariable analysis indicated that metastatic patients were older than non-metastatic patients ($P = 0.024$). In addition, metastatic cases were related to larger tumor size ($P < 0.001$), advanced tumor stage ($P < 0.001$), and higher ISUP/WHO tumor grades ($P = 0.007$), as expected.

Similar to luminal breast cancer samples, a higher frequency of PV-1+ cells among CD-31+ endothelial cells in the primary tumor was observed in metastatic patients compared to non-metastatic patients ($P = 0.032$, Fig. 2A,B). Univariable Cox PH regression analysis was performed for all clinical variables: percentage of PV-1+ cells (per 10-unit increase), tumor size (per 1-cm increase), tumor stage (pT3 vs. pT1-2 as reference), tumor grade (grade III-IV vs. I–II as reference), sex (male as reference), age (per 10-year increase), presence of venous neoplastic thrombosis, and number of comorbidities (Table 5). Among the 52 ccRCC patients, 23 had no comorbidities, while 10 reported a single comorbidity (dyslipidemia $n = 3$; arterial hypertension $n = 5$; Obstructive Sleep Apnea Syndrome (OSAS) $n = 1$; type II diabetes mellitus $n = 1$), and 19 patients had multiple comorbidities. A multivariable Cox PH regression model, after backward stepwise elimination, identified the percentage of PV-1+ cells (HR 1.23, 95% CI 1.02-1.49, $P = 0.028$), tumor size (HR 1.29, 95% CI 1.14–1.46, $P < 0.001$) and tumor grade (HR 3.09, 95% CI 1.27–7.52, $P = 0.013$) as independent prognostic factors for metastasis-free survival (Table 5). The assumptions of the Cox model were met. Time-dependent ROC analysis at 5 years was conducted for each clinical variable. The highest AUC values were observed for tumor size (AUC = 0.875) and PV-1+ cells (AUC = 0.515). Moreover, combining the three variables identified as statistically significant in the multivariable Cox regression model (percentage of PV-1+ cells, tumor size, and tumor grade) resulted in a time-dependent ROC curve with excellent predictive performance (AUC = 0.894). (Table 6; Fig. EV4).

We further investigated PV-1 as a prognostic marker in clear cell renal cell carcinoma (ccRCC) by utilizing publicly available bioinformatic tools and datasets. Specifically, we conducted Kaplan–Meier survival analyses using the KM Plotter tool (Győrffy, 2024) to assess *PLVAP* expression levels (gene level) in ccRCC

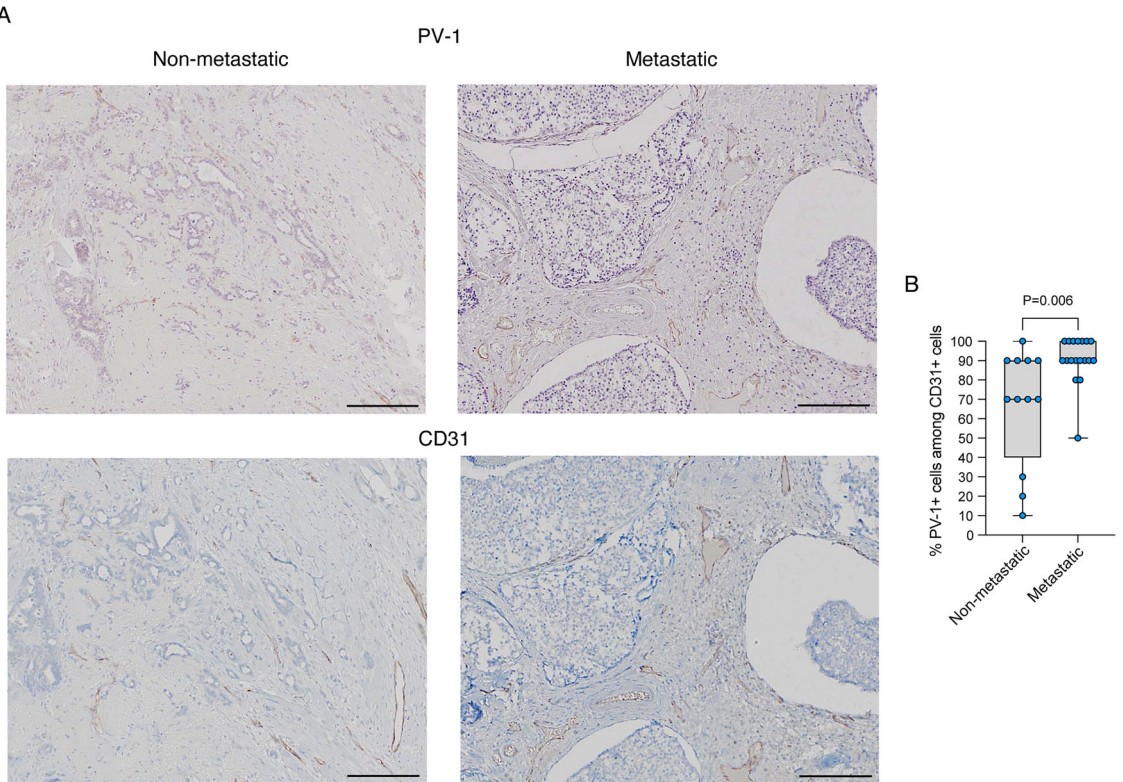

**Figure 1. PV-1+ cell frequency in the primary tumor of metastatic and non-metastatic luminal breast cancer patients.**

(A) Representative images of PV-1 and CD31 staining on FFPE specimens from luminal breast cancer patients with/without metastases. Scale bars, 100 μm. (B) Percentage of PV-1+ cells among CD31+ cells in the primary tumor of metastatic ($n = 18$) and non-metastatic ($n = 12$) luminal breast cancer patients. Data are represented using box-and-whisker plots. Boxplots display values of minimum, first quartile, median, third quartile, and maximum. Each data point represents one sample. Statistical significance was evaluated using the two-sided Mann–Whitney unpaired test. Source data are available online for this figure.

patients. When focusing on mRNA-seq data from Kidney Renal Clear Cell Carcinoma ($n = 117$), we calculated relapse-free survival (RFS). While the $P$ value was not significant, the trend observed was consistent with our findings (Fig. EV2C).

These results suggest that PV-1, reflecting endothelial barrier dysfunction, contributes to the hematogenous spread of renal carcinoma cells. Additional prospective studies are required to evaluate whether the inclusion of PV-1 staining in future stratification models could enhance the prediction of metastatic risk in ccRCC patients. Notably, in many ccRCC patients, PV-1 levels are elevated even in the absence of metastases. Considering that ccRCC is a tumor capable of relapse even decades after the resection of the primary tumor (Abara et al, 2010; Lordan et al, 2008; Nagai et al, 2007; Riviello et al, 2006; Roser et al, 2002), it would be valuable to investigate whether these patients might develop metastases in a longer follow-up period. This would provide important predictive insights into their metastatic potential, helping to identify those who require extended surveillance beyond the standard 5–10 years.

## Soft tissue sarcomas

The cohort included 52 STS patients with different histotypes: 5 dedifferentiated liposarcomas (DDLPS), 10 Gastrointestinal Stromal Tumors (GIST), 6 Retroperitoneal Leiomyosarcomas (LMS), 9

Myxoid LPS (MLPS), 8 Malignant Peripheral Nerve Sheath Tumors (MPNST), 9 Solitary Fibrous Tumors (SFT), and 5 Undifferentiated Pleomorphic Sarcomas (UPS). Main demographics and clinical characteristics of the patients are shown in Appendix Table S1. No significant difference in the frequency of PV-1+ cells among CD-31+ endothelial cells in the primary tumor was observed between metastatic and non-metastatic sarcoma patients (Appendix Fig. S1). Univariable Cox PH regression analysis was performed for all clinical variables: percentage of PV-1+ cells (per 10-unit increase), tumor size (per 1-cm increase), tumor grade (high vs. low as reference), sex (male as reference), age (per 10-year increase). In addition, patients were stratified into high ($n = 27$) and low ($n = 25$) PV-1 groups based on the median value of the frequency of PV1+ cells among CD-31+ endothelial cells (20%) (high vs. low as reference). In the multivariable Cox PH regression model, sex emerged as the sole independent prognostic factors for metastasis-free survival (HR 0.45, 95% CI 0.24–0.98, $P = 0.045$) (Appendix Table S2). The proportional hazards and other model assumptions were adequately met. Then, we considered the possibility that there may be differences according to the analyzed histotype. Thus, we stratified according to histotype and found that PV-1+ cell frequency correlated with clinical outcomes (i.e., metastasis-free survival) in these four specific histotypes, which, notably, are those with the greatest propensity for exclusively hematogenous metastasis: LMS,

**Table 2.  Univariable and multivariable Cox PH regression analyses for luminal breast cancer patients.**

| Variable | Univariable Cox PH model | | | Multivariable Cox PH model | | |
|---|---|---|---|---|---|---|
| | HR | (95% CI) | *P* value | HR | (95% CI) | *P* value |
| % PV-1+ cells[a] | 1.51 | (1.02–2.22) | **0.038** | 1.51 | (1.02–2.22) | **0.038** |
| Tumor stage II | 0.94 | (0.21–4.31) | 0.94 | — | — | — |
| Tumor stage III | 1.25 | (0.25–6.25) | 0.78 | — | — | — |
| ER[a] | 0.98 | (0.30–3.17) | 0.97 | — | — | — |
| PgR[a] | 0.98 | (0.87–1.11) | 0.79 | — | — | — |
| Ki-67[a] | 1.23 | (0.91–1.65) | 0.18 | — | — | — |
| Number of positive lymph nodes | 1.04 | (0.97–1.10) | 0.26 | — | — | — |
| Age[b] | 1.18 | (0.77–1.81) | 0.45 | — | — | — |
| Tumor size[c] | 1.22 | (0.95–1.56) | 0.17 | — | — | — |
| Number of comorbidities | 1.48 | (0.91–2.40) | 0.11 | — | — | — |

*PH* proportional hazards, *HR* hazard ratio, *CI* confidence interval.
The tumor stage is considered a categorical variable taking values I, II, or III (reference: stage I).
[a]Per 10-units increase.
[b]Per 10-years increase.
[c]Per 1-cm increase.
Bold values indicate statistical significance $P \leq 0.05$.

**Table 3.  AUC values from time-dependent ROC curves for luminal breast cancer at 5 years, based on the Kaplan–Meier estimator.**

| Variable | AUC |
|---|---|
| Tumor size | 0.713 |
| % PV-1+ cells | 0.694 |
| Ki-67 | 0.630 |
| Age | 0.514 |
| Number of positive lymph nodes | 0.495 |
| PgR | 0.403 |
| Number of comorbidities | 0.361 |
| ER | 0.315 |
| Tumor stage | 0.315 |

MPNST, SFT, and UPS. Patients classified in the PV-1high group experienced shorter metastasis-free survival compared to those in the PV-1low group (Appendix Fig. S2). We thus unified 27 sarcoma patients (17 of whom had metastases) belonging to these four histotypes. While no significant difference was observed in the frequency of PV-1+ cells among CD-31+ endothelial cells in the primary tumor between metastatic and non-metastatic patients (Fig. EV5A), univariable Cox PH regression analysis was performed for all the clinical variables. These included: the percentage of PV-1+ cells (per 10-unit increase), PV-1 group (high vs. low as reference), tumor size (per 1-cm increase), tumor grade (high vs. low as reference), sex (male as reference), age (per 10-year increase), revealing significant associations (Appendix

Table S3). A multivariable Cox PH regression model identified PV-1 group and tumor size as independent prognostic factors for metastasis-free survival. Specifically, patients in the PV-1high group (% PV-1+/CD31+ cells ≥20) exhibited a hazard ratio of 3.86 (95% CI 1.35-11, *P* = 0.012) compared to those in the PV-1low group (% PV-1+/CD31+ cells <20). Tumor size also emerged as a significant predictor (HR 1.13, 95% CI 1.03-1.24, *P* = 0.008) (Fig. EV5B; Appendix Table S3). The assumptions of the Cox model were adequately met. In addition, a time-dependent ROC analysis at 5 years was conducted for these two variables separately (PV-1 group: AUC = 0.400; tumor size: AUC 0.744) and in combination (AUC = 0.772) (Fig. EV5C). Contrary to luminal breast cancer and ccRCC, the overall analysis of STS did not show a significant difference in PV-1+ cell frequency between metastatic and non-metastatic patients. However, upon stratifying by histotype, significant findings emerged for specific subtypes: LMS, MPNST, SFT, and UPS. In these subtypes, patients with more than 20% of PV-1+ cells exhibited markedly shorter metastasis-free survival, indicating that PV-1's prognostic value might be histotype-specific within the heterogeneous group of STS. Cox PH regression models identified high PV-1+ cell frequency and larger tumor size as significant prognostic factors associated with poor clinical outcomes, suggesting that endothelial permeability, as reflected by PV-1 detection, may play a critical role in the metastatic process for these sarcoma subtypes. While these findings on STS are encouraging and provide valuable insight, they remain preliminary due to the cohort's limited size and heterogeneity, warranting further validation.

Our study identifies PV-1 as a potential prognostic marker associated with distant metastases in luminal breast cancer and ccRCC. While our findings did not directly assess endothelial permeability, we employed PV-1 expression as a proxy for increased vascular permeability based on previous evidence from animal models and human observations. In murine studies, PV-1 was upregulated in intestinal vasculature following oral administration of *Salmonella*, and the bacteria were subsequently detected in the liver. Notably, in a genetic mouse model with endothelial-specific β-catenin gain-of-function, which impairs PV-1 upregulation, *Salmonella* failed to reach the liver, implicating PV-1 in endothelial barrier modulation (Spadoni et al, 2015). Similarly, high-fat diet feeding led to increased PV-1 expression, which correlated with elevated intestinal permeability, as shown by FITC-dextran translocation into the bloodstream and intravital imaging using CELLVIZIO endoscopy (Mouries et al, 2019; Sorribas et al, 2019). Elevated PV-1 levels have also been reported in NASH patients, reinforcing its association with endothelial dysfunction in human disease (Mouries et al, 2019). Furthermore, in a colorectal cancer (CRC) mouse model, increased PV-1 expression was associated with enhanced vascular permeability and translocation of bacteria and tumor cells into the liver (Bertocchi et al, 2021). Nonetheless, we acknowledge that we have not functionally demonstrated the role of PV-1 in barrier dysfunction in the analyzed tumors. Our findings should be interpreted cautiously, given the retrospective design and limited sample size. Future research should focus on prospective validation in larger cohorts and include direct functional assays to clarify the mechanistic role of PV-1 in cancer dissemination, as well as to explore the therapeutic potential of targeting of PV-1-mediated pathways to improve patient outcomes.

**Table 4. Demographics and clinical characteristics of the ccRCC patients (n = 52).**

| n | | Overall 52 | Non-metastatic 30 | Metastatic 22 | P value | Test |
|---|---|---|---|---|---|---|
| Female sex, n (%) | | 19 (36.5) | 13 (43.3) | 6 (27.3) | 0.370 | Chi-squared |
| Age, median [Q1, Q3] | | 64.0 [58.0, 73.0] | 68.5 [63.2, 74.8] | 59.0 [54.2, 69.2] | **0.024** | Mann–Whitney |
| Percentage of PV-1+ cells, median [Q1, Q3] | | 90.0 [60.0, 100.0] | 75.0 [50.0, 97.5] | 100.0 [72.5, 100.0] | **0.032** | Mann–Whitney |
| Follow-up time or metastasis onset (months), median [Q1, Q3]* | | 87.1 [9.5, 189.7] | 185.4 [124.6, 197.4] | 4.5 [0.0, 15.0] | **<0.001** | Mann–Whitney |
| Tumor size (mm), median [Q1, Q3] | | 46.5 [30.0, 80.0] | 32.5 [25.0, 45.8] | 80.0 [55.0, 105.0] | **<0.001** | Mann–Whitney |
| Presence of venous neoplastic thrombi, n (%) | | 12 (23.1) | 4 (13.3) | 8 (36.4) | 0.106 | Chi-squared |
| Tumor location, n (%) | left kidney | 23 (44.2) | 12 (40.0) | 11 (50.0) | 0.664 | Chi-squared |
| | right kidney | 29 (55.8) | 18 (60.0) | 11 (50.0) | | |
| Tumor stage, n (%) | 1a | 21 (40.4) | 19 (63.3) | 2 (9.1) | **<0.001** | Chi-squared |
| | 1b | 11 (21.2) | 6 (20.0) | 5 (22.7) | | |
| | 2 | 4 (7.7) | | 4 (18.2) | | |
| | 3a | 4 (7.7) | | 4 (18.2) | | |
| | 3b | 11 (21.2) | 4 (13.3) | 7 (31.8) | | |
| | 3c | 1 (1.9) | 1 (3.3) | | | |
| Tumor grade, n (%) | I | 7 (14.0) | 7 (24.1) | | **0.007** | Chi-squared |
| | II | 30 (60.0) | 19 (65.5) | 11 (52.4) | | |
| | III | 12 (24.0) | 3 (10.3) | 9 (42.9) | | |
| | IV | 1 (2.0) | | 1 (4.8) | | |

*Follow-up time refers to non-metastatic patients, while metastasis onset refers to metastatic patients. There were 2 missing data for the tumor size, which were imputed with the mean value of the other patients. There were 2 missing data for the tumor grade, which were imputed with the median value of the other patients, i.e., with grade II.
Bold values indicate statistical significance P ≤ 0.05.

# Methods

### Reagents and tools table

| Reagent/resource | Reference or source | Identifier or catalog number |
|---|---|---|
| **Experimental models** | | |
| **Recombinant DNA** | | |
| **Antibodies** | | |
| Monoclonal Mouse Anti-Human CD31 | Agilent | Cat #M082329-2 |
| Polyclonal Rabbit Anti-Human PLVAP | Sigma-Aldrich | Cat # HPA002279 |
| **Oligonucleotides and other sequence-based reagents** | | |
| **Chemicals, enzymes, and other reagents** | | |
| BOND Polymer Refine Detection Kit | Leica Biosystems | Cat #DS9800 |
| BOND Epitope Retrieval Solution 1 | Leica Biosystems | Cat #AR9961 |
| BOND Primary Antibody Diluent | Leica Biosystems | Cat #AR9352 |
| **Software** | | |
| Fiji/ImageJ | http://fiji.sc/ | RRID: SCR_002285 |

| Reagent/resource | Reference or source | Identifier or catalog number |
|---|---|---|
| **Other** | | |
| BOND-MAX Automated Immunohistochemistry Vision Biosystem | Leica Biosystems | |
| Olympus BX61 Upright Microscope | Olympus | |

## Human samples

Formalin-fixed paraffin-embedded (FFPE) samples of the primary tumor of 30 luminal breast cancer patients were retrieved from the archives of the Pathology Department of Humanitas Research Hospital. These patients were enrolled into a retrospective observational study (ICH/MIM/002 "Identification of the Breast Vascular Barrier and its role in breast cancer metastatization process"). All patients underwent resection of primary breast cancer and adjuvant endocrine therapy. Endocrine therapy included aromatase inhibitors and tamoxifen plus GnRH analog in premenopausal women. Twenty-one patients (14 and 7 among the metastatic and non-metastatic cohorts, respectively) also received chemotherapy containing anthracylines and taxanes. 18 patients developed distant metastases within 2 years of initiating

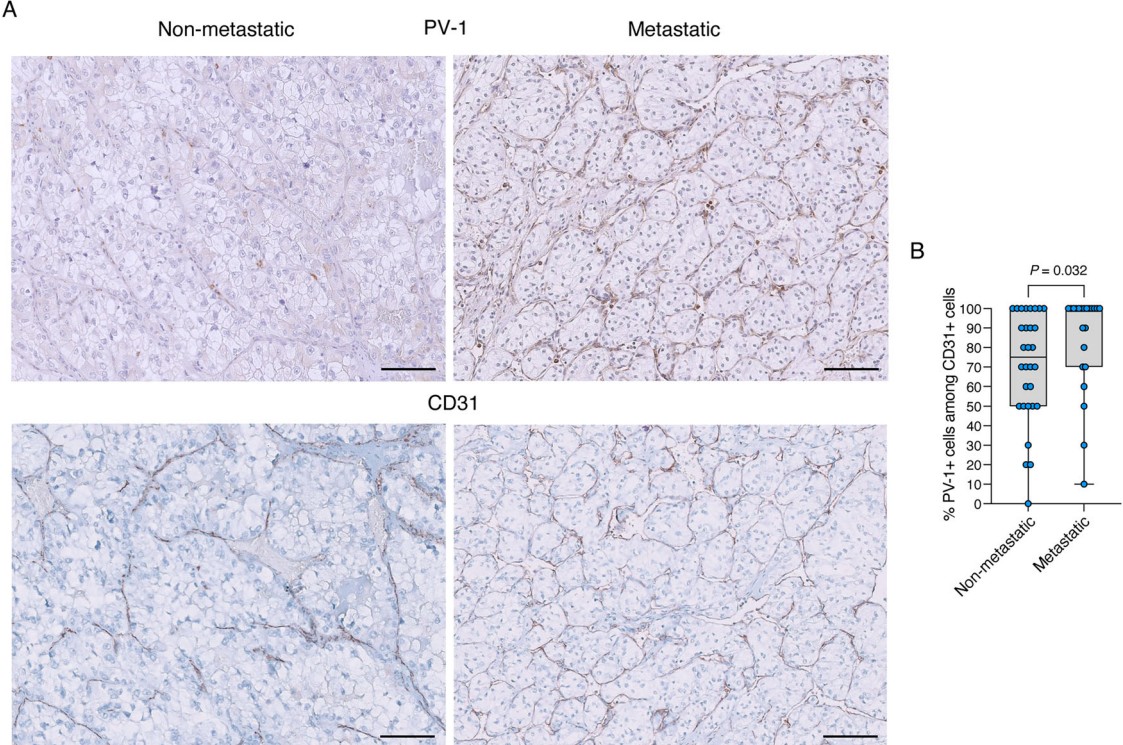

**Figure 2. PV-1+ cell frequency in the primary tumor of metastatic and non-metastatic ccRCC patients.**

(A) Representative images of PV-1 and CD31 staining on FFPE specimens from ccRCC patients with/without metastases. Scale bars, 100 µm. (B) Percentage of PV-1+ cells among CD31+ cells in the primary tumor of metastatic (n = 22) and non-metastatic (n = 30) ccRCC patients. Data are represented using box-and-whisker plots. Boxplots display values of minimum, first quartile, median, third quartile, and maximum. Each data point represents one sample. Statistical significance was evaluated using the two-sided Mann–Whitney unpaired test. Source data are available online for this figure.

**Table 5. Univariable and multivariable Cox PH regression analyses for ccRCC patients.**

| Variable | Univariable Cox PH model | | | Multivariable Cox PH model | | |
|---|---|---|---|---|---|---|
| | HR | (95% CI) | P value | HR | (95% CI) | P value |
| % PV-1+ cells[a] | 1.15 | (0.96–1.39) | 0.14 | 1.23 | (1.02–1.49) | **0.028** |
| Tumor size[b] | 1.26 | (1.14–1.40) | **<0.001** | 1.29 | (1.14–1.46) | **<0.001** |
| Tumor stage[c] | 3.04 | (1.31–7.06) | **0.010** | — | — | — |
| Tumor grade[d] | 4.15 | (1.76–9.77) | **0.001** | 3.09 | (1.27–7.52) | **0.013** |
| Female sex | 0.59 | (0.23–1.52) | 0.28 | — | — | — |
| Age[e] | 0.74 | (0.56–0.97) | **0.027** | — | — | — |
| Number of comorbidities | 0.79 | (0.54–1.17) | 0.25 | — | — | — |
| Presence of venous neoplastic thrombosis | 2.36 | (0.98–5.64) | 0.054 | — | — | — |

*PH* proportional hazards, *HR* hazard ratio, *CI* confidence interval.
[a]Per 10-unit increase.
[b]Per 1-cm increase.
[c]pT3 vs. pT1-2 (reference).
[d]III-IV vs. I–II (reference).
[e]Per 10-year increase.
Bold values indicate statistical significance P ≤ 0.05.

**Table 6. AUC values from time-dependent ROC curves for ccRCC at 5 years, based on the Kaplan–Meier estimator.**

| Variable | AUC |
|---|---|
| Tumor size + Tumor grade + % PV-1+ cells | 0.894 |
| Tumor size | 0.875 |
| % PV-1+ cells | 0.515 |
| Tumor stage | 0.457 |
| Tumor grade | 0.447 |
| Presence of venous neoplastic thrombosis | 0.345 |
| Age | 0.322 |
| Number of comorbidities | 0.175 |
| Sex | 0.138 |

treatment, whereas 12 patients did not develop distant metastases within 5 years. Formalin-fixed paraffin-embedded (FFPE) samples of the primary tumor of 52 ccRCC patients were retrieved from the archives of the Pathology Department of Humanitas Research Hospital. These patients were enrolled in a retrospective observational study (ICH 40/20 "Retrospective Analysis of the Expression of a Novel Molecular Marker Associated with Renal Endothelial Functionality in Predicting Distant Metastasis in Patients with Clear Cell Renal Carcinoma"). A total of 22 patients developed distant metastases at the time of the diagnosis or during the follow-

up, whereas 30 patients did not develop distant metastases within 15 years of follow-up. All these patients underwent either partial or radical nephrectomy. After surgery, 11 patients received chemotherapy. Formalin-fixed paraffin-embedded (FFPE) samples of the primary tumor of 52 soft tissue sarcoma patients were retrieved from the Humanitas Research Hospital pathology laboratory. These patients were enrolled into a retrospective observational study ("Vessels encapsulating tumor clusters (VETC), prognostic and predictive value in Renal Cell Carcinoma, Adrenal Gland Carcinoma and Sarcoma"). All analyzed patients underwent surgery with curative intent for primary non-metastatic STS (i.e., no synchronous metastases were present at the index operation). After surgery 12 patients received adjuvant chemotherapy, radiotherapy or both. For the purpose of this study, the patients were divided into two groups: 28 patients developing metachronous distant metastases during the follow-up; 24 non-metastatic patients who did not experience any distant metastases after a minimum follow-up of 10 years. Age and sex information were collected and incorporated into the analyses, in compliance with applicable privacy constraints. All these studies were conducted at Istituto Clinico Humanitas and approved by the institutional review board of Istituto Clinico Humanitas. All participants signed an informed consent, and the experiments conformed to the principles set out in the WMA Declaration of Helsinki and the Department of Health and Human Services Belmont Report.

## Immunohistochemistry

Immunohistochemical analysis of tumor samples was performed on FFPE tissue sections. Immunohistochemical staining was performed in a Bond-Max Automated Immunohistochemistry Vision Biosystem (Leica Microsystems GmbH, Wetzlar, Germany) using the Bond Polymer Refine Detection Kit (DS9800). 3-mm-thick sections were prepared from FFPE human cancer tissue blocks, deparaffinized, pretreated with the Epitope Retrieval Solution 1 (pH 6) at 100 °C (20 min for CD31, 40 min for PV-1) and then incubated for 30 min with primary antibody anti-human CD31 (DAKO, #M0823, final concentration = 6.8 µg/ml), anti-human PV-1 (Sigma-Aldrich, # HPA002279 1:200) antibodies diluted in Bond Primary Antibody Diluent for PV-1 and CD31 (AR9352). Serial sections of the same FFPE sample were stained for PV-1 and CD31, respectively.

## Analysis of PV-1 protein detection on human samples

All tissue sections stained for PV-1 and CD31 were scored blindly by the pathologist. The analyses consisted of the evaluation of the percentage of PV-1+ cells among the CD31+ vessels through an optical microscope (Olympus) at a ×100 field as indicated in Bertocchi et al (Bertocchi et al, 2021). From each patient, one slide of a representative tumor area, including well-recognizable vascularity under study, was selected.

The association between PV-1 detection and metastasis-free survival was tested by a third investigator (RS), who did not participate in the scoring process.

## Kaplan–Meier survival analyses using the KM Plotter tool

Kaplan–Meier survival analyses were performed using the KM Plotter tool (Győrffy, 2021) to assess the impact of PV-1 expression

levels in luminal A and B breast cancer as well as in ccRCC. For luminal breast cancer, the breast cancer mRNA gene chip datasets were used, and the probe *PLVAP* (221529_s_at) was selected. The PAM50 subtype classification was applied to select luminal A or luminal B patients, and the analysis was restricted to those who had received endocrine therapy. Kaplan–Meier curves were generated for distant metastasis-free survival (DMFS) and relapse-free survival (RFS). The survival data were derived from multiple datasets as follows: Luminal A: DMFS from 8 datasets (total $n = 313$); RFS from 13 datasets (total $n = 481$). Luminal B: DMFS from 7 datasets (total $n = 200$); RFS from 12 datasets (total $n = 321$). To evaluate PV-1 protein expression in breast cancer, the KM plotter for protein expression in breast tumors was used. Only one dataset was available for this analysis (Tang et al, 2018), comprising 65 tumor samples. This tool allowed for the calculation of overall survival (OS) only, and it did not permit the specification of molecular subtypes. For ccRCC, the pan-cancer mRNA RNA-seq dataset was used, selecting kidney renal clear cell carcinoma and *PLVAP* gene. An RFS Kaplan–Meier curve was generated ($n = 117$). In all analyses, patients were stratified into high and low PV-1 expression groups using an automatic cut-off: all possible cut-off values between the lower and upper quartiles were tested, and the best-performing threshold was selected. The False Discovery Rate (FDR) was reported alongside the $P$ value.

## ROC plotter tool

The ROC Plotter tool was used to evaluate *PLVAP* as a prognostic biomarker in breast cancer. ROC Plotter performs a meta-analysis by integrating major publicly available datasets annotated with tumor subtypes, treatment, and relapse-free survival (RFS) data, treating them as a unified cohort (Fekete and Győrffy, 2019). The tool constructs a ROC curve using 5-year RFS as the clinical endpoint. Patients are classified based on clinical outcome: those without relapse within 5 years and those who experienced relapse within 5 years. Gene expression is evaluated as a continuous predictor of outcome. The analysis was performed using the *PLVAP* probe (221529_s_at), selecting patients classified as luminal A or luminal B and treated with endocrine therapy.

## Statistical analysis

Normality of the data was assessed using the Shapiro–Wilk test. Comparison of two groups of continuous data was performed using the two-sided $t$ test for normally distributed data and the two-sided Mann–Whitney test for non-normally distributed data. Categorical data were compared using the chi-squared test, or the Fisher's exact test when any group contained fewer than five observations. Missing data were imputed using the mean value of the known data for all patients in each tumor type. All statistical analyses were conducted in Python, version 3.11.7. Demographic tables and Cox models were generated using the tableone and lifelines packages, respectively. Statistical significance was set at $P < 0.05$.

Univariable Cox proportional hazards (PH) regression was performed for all clinical variables, followed by a backward stepwise elimination process to identify independent prognostic factors. Variables with a $P$ value $< 0.20$ in the univariable Cox regression were considered eligible for inclusion in the backward elimination process. This relaxed threshold was chosen to ensure that

**The paper explained**

**Problem**

Predicting which cancer patients will develop distant metastases remains difficult, especially in tumors that spread through the bloodstream. New biomarkers are urgently needed to improve risk assessment and guide treatment.

**Results**

We identified PV-1, a protein linked to blood vessel permeability, as a new prognostic marker for metastasis in luminal breast cancer and clear cell renal cell carcinoma (ccRCC). High PV-1 levels in tumors predicted shorter metastasis-free survival, even beyond standard clinical factors. Public datasets confirmed that high *PLVAP* gene expression was linked to worse outcomes. Preliminary data also suggest a role for PV-1 in select sarcoma subtypes.

**Impact**

PV-1 could help identify patients at high risk of metastasis at diagnosis, enabling more tailored follow-up and treatment. If validated, PV-1 testing could become part of routine cancer care to support personalized management.

potentially relevant predictors were not prematurely excluded. Time-dependent ROC curves were generated with Stata version 18 using the stroccurve package. In ccRCC and in STS, to create the time-dependent ROC curve for the combination of the statistically significant variables of the multivariable Cox regression model (percentage of PV1+ cells, tumor size, and tumor grade in ccRCC and PV-1 group and tumor size in STS), linear predictors (risk scores) for each patient were calculated, and then a time-dependent ROC was generated using this score. To generate the time-dependent ROC curve for the tumor stage in luminal breast cancer, we created dummy variables and combined stage information into a risk score. Then the ROC curve was generated using this score. In both tumors, ROC curves may appear discretized (step-like) due to the small sample size or the categorical nature of some variables, thus AUC estimates should be interpreted with caution.

## Data availability

This study includes no data deposited in external repositories.

The source data of this paper are collected in the following database record: biostudies:S-SCDT-10_1038-S44321-025-00277-5.

## Peer review information

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

## Acknowledgements

The research leading to these results has received funding from AIRC under the 5 per Mille 2018 - ID.21147 program and AIRC under the 5 per Mille 2019 - ID. 22757 program. Visual abstract created in https://BioRender.com.

## Author contributions

**Chiara Pozzi**: Data curation; Formal analysis; Writing—original draft; Project administration; Writing—review and editing. **Riccardo Sarti**: Data curation; Formal analysis; Methodology; Writing—original draft; Writing—review and editing. **Antonino Lo Cascio**: Data curation; Investigation; Methodology; Project administration. **Stefanos Bonovas**: Formal analysis; Supervision; Methodology; Writing—review and editing. **Luca Tiraboschi**: Data curation; Formal analysis; Methodology. **Michela Lizier**: Data curation; Formal analysis; Methodology; Writing—review and editing. **Alice Bertocchi**: Conceptualization; Data curation; Formal analysis; Writing—review and editing. **Rosalba Torrisi**: Conceptualization; Data curation; Supervision; Investigation. **Bethania Fernandes**: Formal analysis; Methodology. **Piergiuseppe Colombo**: Conceptualization; Data curation; Formal analysis; Supervision; Methodology; Writing—original draft. **Pietro Diana**: Data curation. **Emilia Maria Cristina Mazza**: Formal analysis; Validation; Methodology; Writing—review and editing. **Marina Valeri**: Formal analysis. **Salvatore Lorenzo Renne**: Data curation; Formal analysis; Methodology; Writing—original draft. **Ferdinando Carlo Maria Cananzi**: Conceptualization; Data curation; Writing—original draft. **Laura Samà**: Formal analysis. **Alexia Bertuzzi**: Data curation. **Corrado Tinterri**: Supervision. **Massimo Lazzeri**: Data curation; Writing—review and editing. **Maria Rescigno**: Conceptualization; Data curation; Supervision; Funding acquisition; Writing—original draft; Writing—review and editing.

Source data underlying figure panels in this paper may have individual authorship assigned. Where available, figure panel/source data authorship is listed in the following database record: biostudies:S-SCDT-10_1038-S44321-025-00277-5.

## Disclosure and competing interests statement

Maria Rescigno is an EMM editorial board member, *EMBO* Member and *EMBO* Council Chair. This has no bearing on the editorial consideration of this article for publication. The remaining authors declare no competing interests.

# Expanded View Figures

**Figure EV1.   Time dependent ROC curves for luminal breast cancer at 5 years.**

Plots of the time-dependent ROC analysis at 5 years performed for each of the clinical variables analyzed: tumor size, percentage of PV-1+ cells among CD31+ cells, Ki-67, age, number of positive lymph nodes, PgR, number of comorbidities, ER and tumor stage. $n = 30$ (metastatic, $n = 18$; non-metastatic, $n = 12$).

▶

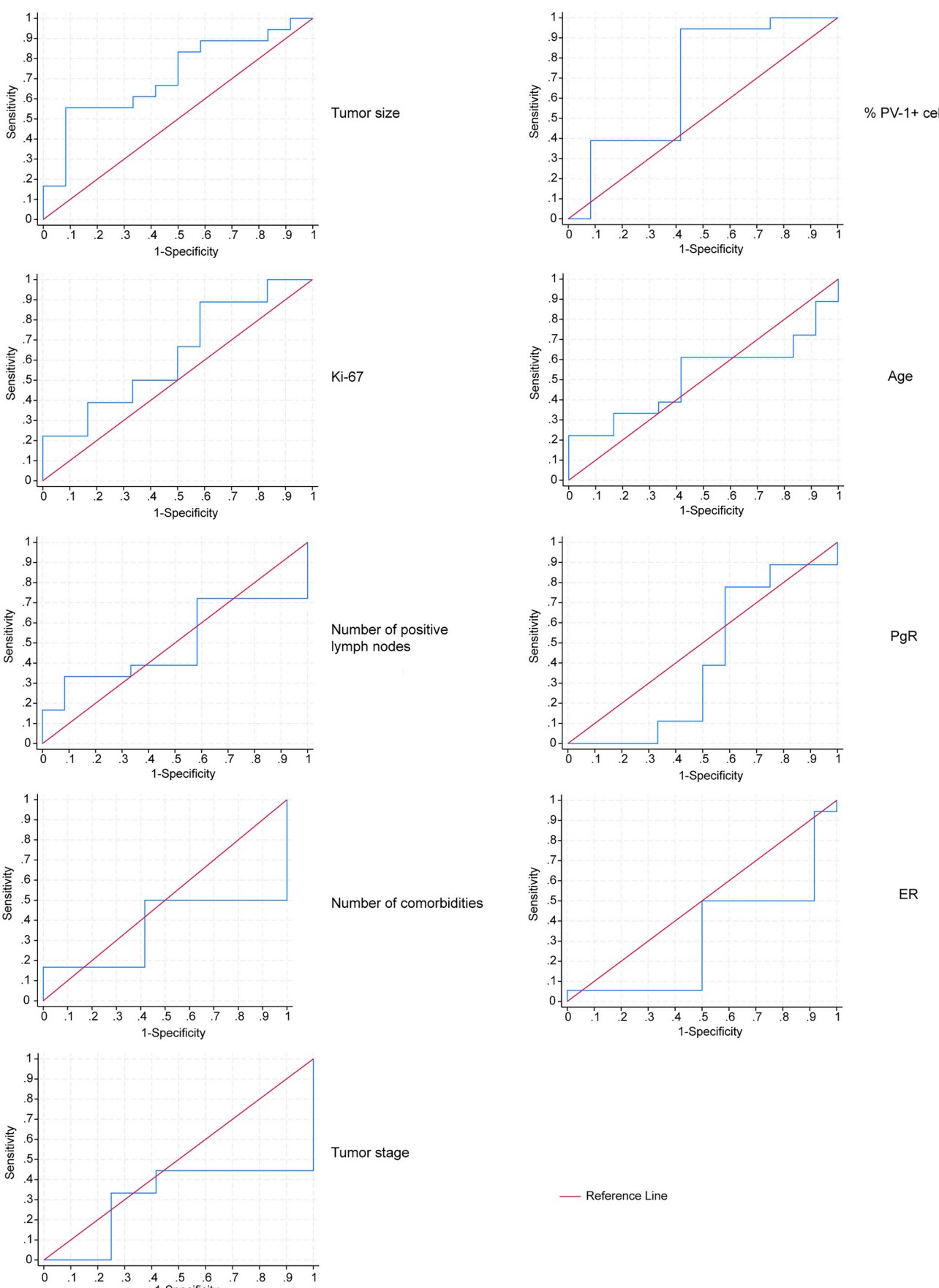

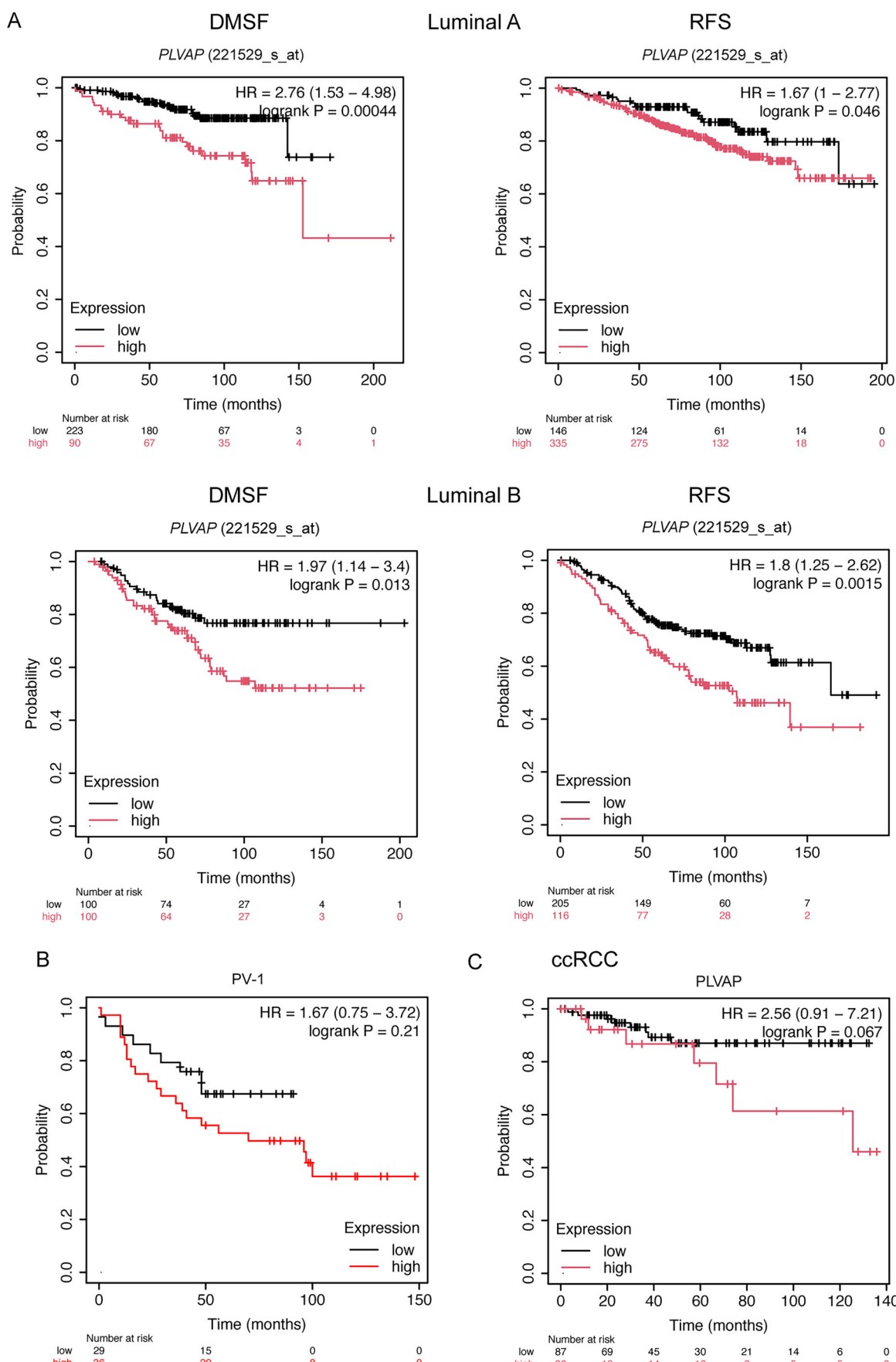

◄ **Figure EV2. Kaplan–Meier survival analysis of PV-1 expression levels using the KM plotter tool.**

(A) Distant-metastasis-free survival (DMSF) and relapse-free survival (RFS) were analyzed using mRNA gene chip data from Luminal A (DMSF, $n = 313$; RFS, $n = 481$) and Luminal B (DMSF, $n = 200$; RFS, $n = 321$) breast cancer patients treated with endocrine therapy. (B) Overall survival (OS) analysis based on PV-1 protein expression in breast cancer tissue ($n = 65$), including Luminal A, Luminal B, HER2-positive, and triple-negative breast cancer (TNBC) subtypes. (C) Relapse-free survival (RFS) was analyzed using mRNA-seq data from ccRCC patients ($n = 117$).

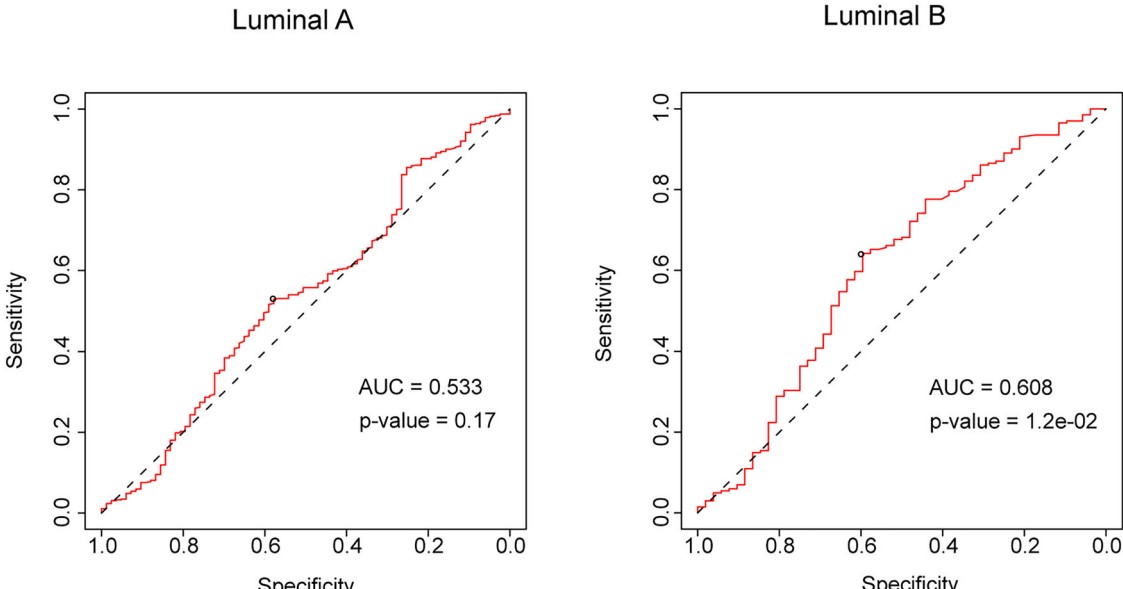

**Figure EV3. ROC curve analysis of PV-1 expression in luminal breast cancer subtypes using ROC Plotter.**

ROC Plotter constructs the ROC curve using 5-year relapse-free survival (RFS) as the clinical endpoint. Patients are classified based on clinical outcome: those without relapse within 5 years, and those with relapse within 5 years. PV-1 gene expression is evaluated as a continuous predictor of outcome. The analysis was performed separately for luminal A ($n = 637$; AUC $= 0.533$) and luminal B ($n = 253$; AUC $= 0.608$) breast cancer patients treated with endocrine therapy.

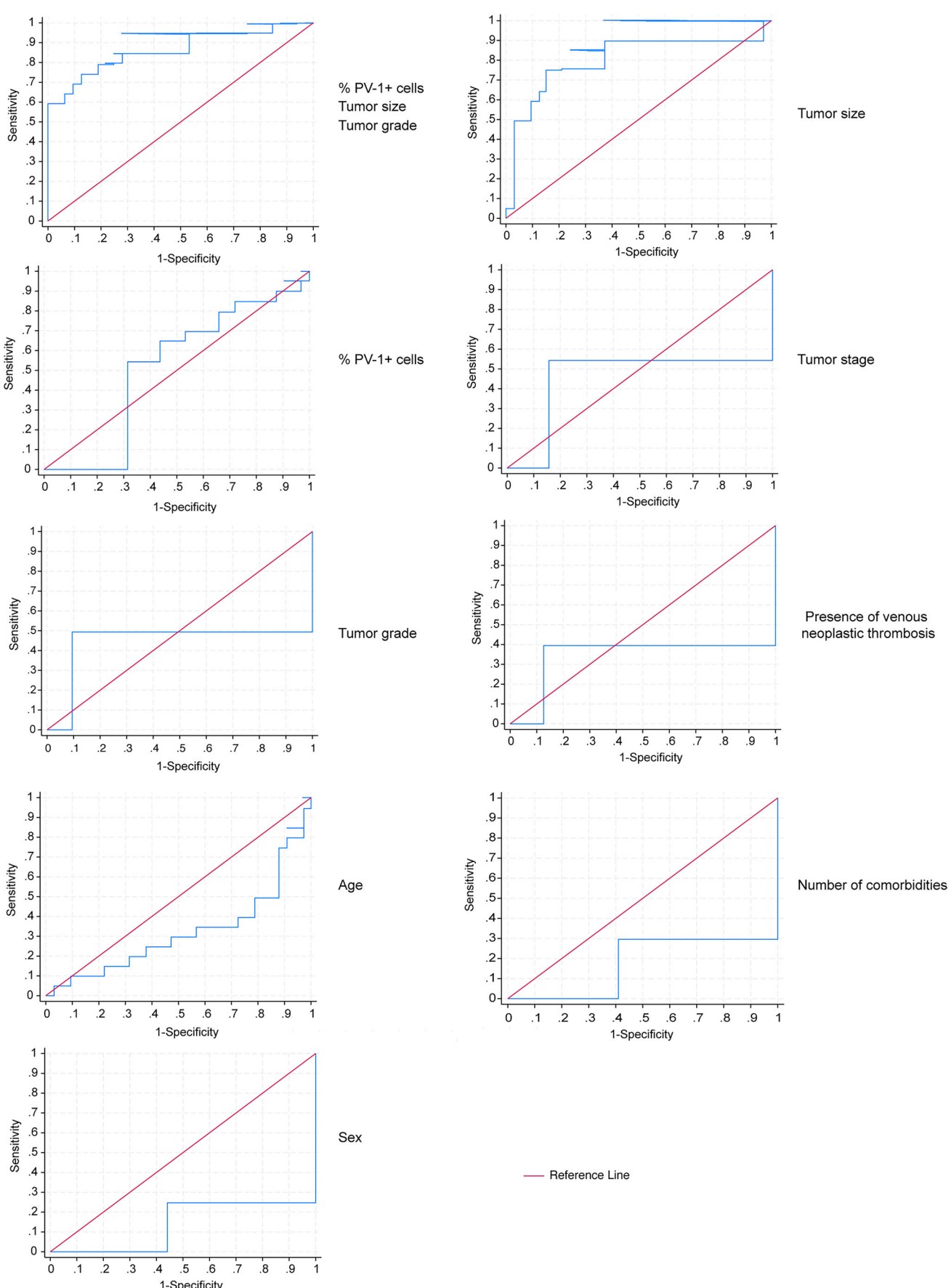

◀ **Figure EV4. Time-dependent ROC curves for ccRCC at 5 years.**

Plots of the time-dependent ROC analysis at 5 years performed for each of the clinical variables analyzed (tumor size, percentage of PV-1+ cells among CD31+ cells, tumor stage, tumor grade, presence of venous neoplastic thrombosis, age, number of comorbidities and sex) and for the combination of tumor size, tumor grade, and percentage of PV-1+ cells. $n = 52$ (metastatic, $n = 22$; non-metastatic, $n = 30$).

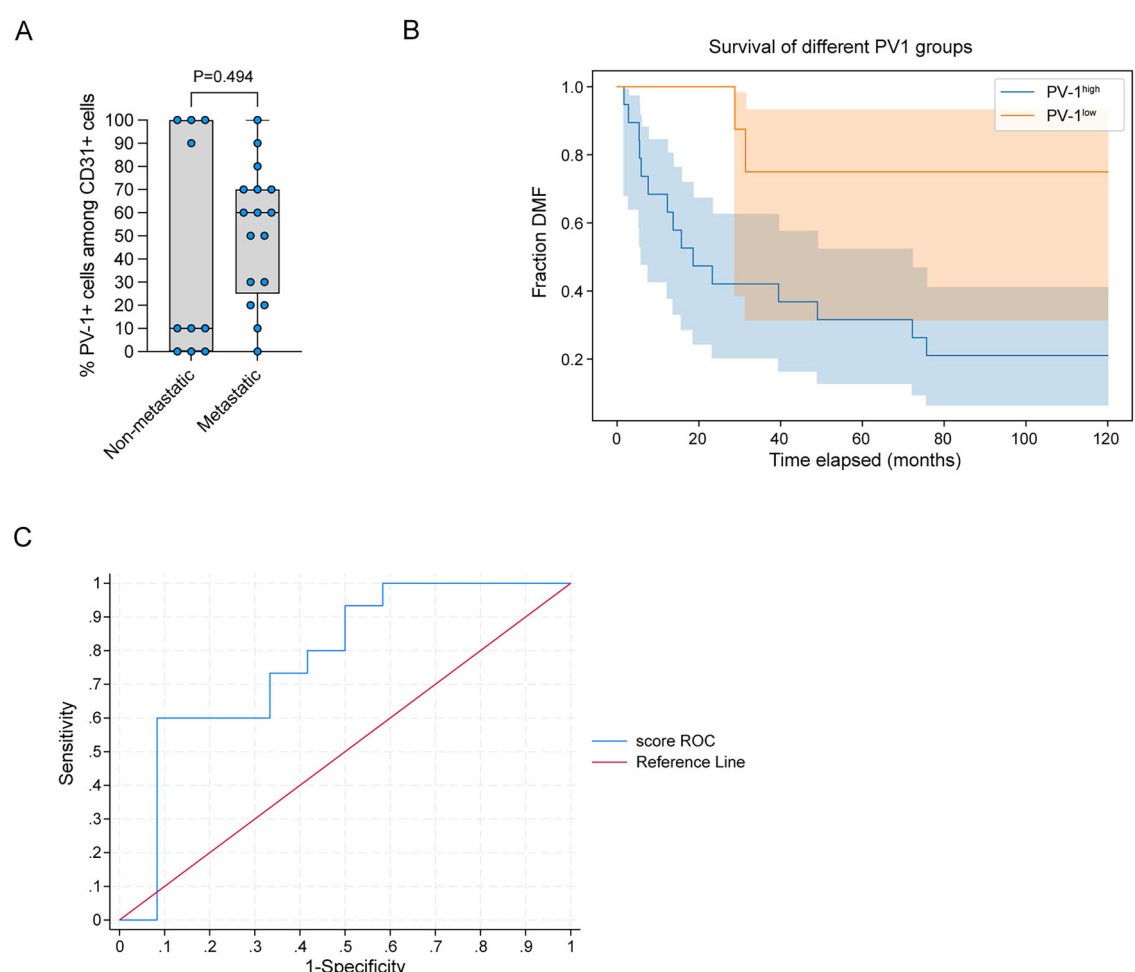

**Figure EV5. PV-1+ cell frequency in the primary tumor of metastatic and non-metastatic sarcoma patients from the LMS, MPNST, SFT, and UPS histotypes.**

(A) Percentage of PV-1+ cells among CD31+ cells in the primary tumor of metastatic ($n = 17$) and non-metastatic ($n = 10$) sarcoma patients. Data are represented using box and whisker plots. Boxplots display values of minimum, first quartile, median, third quartile, and maximum. Each data point represents one sample. Statistical significance was evaluated using the two-sided Mann–Whitney unpaired test. (B) Metastasis-free survival of sarcoma patients depending on the PV-1 group (high vs. low). PV-1high group: % PV-1+/CD31+ cells ≥20; PV-1low: % PV-1+/CD31+ cells <20. $n = 27$ (metastatic, $n = 17$; non-metastatic, $n = 10$). (C) Plot of the time-dependent ROC analysis at 5 years performed for the combination of PV-1 group and tumor size (score). $n = 27$ (metastatic, $n = 17$; non-metastatic, $n = 10$).

