## [Peer Review File · EMBO Molecular Medicine]

PV-1: A Novel Molecular Prognostic Marker of Distant Metastases in Various Solid Tumors

Chiara Pozzi, Riccardo Sarti, Antonino Lo Cascio, Stefanos Bonovas, Luca Tiraboschi, Michela Lizier, Alice Bertocchi, Rosalba Torrisi, Bethania Fernandes, Piergiuseppe Colombo, Pietro Diana, Emilia Mazza, Marina Valeri, Salvatore Renne, Ferdinando Cananzi, Laura Sama', Alexia Bertuzzi, Corrado Tinterri, Massimo Lazzeri, and Maria Rescigno

Corresponding author: Maria Rescigno (maria.rescigno@hunimed.eu)

Review Timeline:

Submission Date:	24th Jul 24
Editorial Decision:	4th Sep 24
Revision Received:	11th Feb 25
Editorial Decision:	7th Mar 25
Revision Received:	13th Jun 25
Editorial Decision:	2nd Jul 25
Revision Received:	6th Jul 25
Accepted:	9th Jul 25

Editor: Lise Roth

Transaction Report:

4th Sep 2024

Dear Maria,

Thank you for the submission of your manuscript to EMBO Molecular Medicine, and please accept my apologies for the delay in getting back to you in this busy time of the year. We have now received feedback from the three reviewers who agreed to evaluate your manuscript.

As you will see from the enclosed reports, the referees acknowledge the potential translational interest of the findings, however referees #2 and #3 also raise several concerns regarding the low sample number and sample selection criteria, the lack of proper standards for prognostic/predictive studies, potential confounding factors, etc.

Based on the nature of the concerns and considering that at EMBO Press we encourage a single round of revisions in a limited time frame, we prefer to return the manuscript to you at this point with the decision that we cannot offer to publish it.

Given the potential interest of the findings, we would, however, be willing to consider a new manuscript on the same topic if at some time in the near future you obtained data that would considerably strengthen the message of the study and particularly the biomarker aspect (i.e. functional studies, referee #3 point #8, would not be required). To be completely clear, however, I would like to stress that if you were to send a new manuscript, this would be treated as a new submission rather than a revision and would be reviewed afresh, in particular with respect to the literature and the novelty of your findings at the time of resubmission. If you decide to follow this route, please make sure you nevertheless upload a letter of response to the referees' comments.

I am sorry that I could not bring better news this time and hope that the referee comments are helpful in your continued work in this area.

With kind regards,

Lise

***** Reviewer's comments *****

Referee #1 (Remarks for Author):

This is a well-conducted and highly interesting study that analyzes the potential role of PV-1 as a marker for distant metastasis in various solid tumors. The study presents a clear hypothesis, utilizes appropriate laboratory and statistical methodologies, and is well-written and clearly presented.

PV-1 is known as a marker of endothelial permeability, and the authors have previously demonstrated its involvement in the dissemination of colorectal cancer. In this study, they show that the frequency of PV-1-positive cells is a predictor of metastasis-free survival in luminal breast cancer, clear cell renal cell carcinoma (ccRCC), and certain histotypes of soft tissue sarcomas. The manuscript offers compelling findings that, if validated in future studies, could significantly aid clinicians and enhance diagnostic and therapeutic strategies.

I have only a few minor comments:

1. What do the authors think about the possibility of assessing blood levels of PV-1? Although blood levels do not provide certainty about the protein's origin, it would be interesting to evaluate this in future prospective studies due to the simplicity of blood sampling. I would appreciate the authors' thoughts on this matter.
2. The results are of great interest for all three types of tumors, although, as mentioned by the authors in the discussion, they require validation. However, in my opinion, the results on soft tissue tumors are somewhat less robust due to the small sample size related to the subtype. I recommend reducing the emphasis on the findings related to this type of tumor.
3. Line 21: The authors reported "cell renal cell carcinoma" instead of "clear cell renal cell carcinoma."

Referee #2 (Comments on Novelty/Model System for Author):

Elevated levels of PV-1 in the primary tumor have been previously shown to double the risk of developing distant metastasis in colorectal cancer patients. In this study, the frequency of PV-1 in endothelial cells was identified as a potential prognostic marker for distant metastases and metastasis free survival in luminal breast cancer, clear cell renal cell carcinoma and in histotype stratified soft tissue sarcomas. The presented results are novel and indicate that analysis of PV-1 expression may provide prognostic information. Therefore, this study is of clinical interest. However, further validation is required to confirm these findings and the title of the manuscript should be modified accordingly.

The major shortcoming of this study is the relatively low number of samples analysed. Especially in histotype stratified soft tissue sarcomas, the only one sample was representing non-metastatic cancers in LMS and UPS groups and therefore, these results should be considered as very preliminary rather than as compelling evidence that PV-1 is a prognostic marker of distant metastases in these STS subtypes.

In addition, it is unclear why the number of axillary lymph nodes having cancer cells was not presented, as it is commonly used as a prognostic indicator in breast cancer. Therefore, the advantage of analyzing PV-1 biomarker expression instead of the number of lymph nodes with cancer cells is not clear. For example, can PV-1 expression levels in endothelial cells of primary tumors identify those 30 % of patients without lymph node involvement at diagnosis who have increased risk of disease relapse and distant metastases? Also, it is not fully clear what advantage the analysis PV-1 expression provide over measurement of tumor size or tumor grade in ccRCC.

At present, there is a lot of detailed information about metastatic processes for example in luminal breast cancer and ccRCC and therefore, it is an oversimplification to conclude that mechanism underlying dissemination remain poorly understood (Introduction, page 3, lines 68 and 69 as well as 83-84). The authors should state more specifically which dissemination processes they are referring to and provide references to support the conclusion.

Referee #3 (Remarks for Author):

The authors have previously shown that PV-1 may be a prognostic marker for distant metastasis in CRC. The present work is an extension of this earlier work in other solid tumors such as luminal BC, ccRC and in STS. The authors employed 134 human tumor tissue samples to examine the frequency of PV-1+ endothelial cells and its association with metastatic status. However, the manuscript heavily relies on limited FFPE staining and statistical analysis of %PV-1+/CD31+ cells and survival analysis, which raises significant concerns about the robustness of the findings. In order to further develop their work, the authors are encouraged to consider the following:

1. The most critical conceptual limitation of the study is that the authors did not apply proper standards of a prognostic/predictive study. Biomarkers are not to be validated by comparing cohorts. The terms "prognostic markers" and "predictive markers" apply to individuals and not to populations. Population analyses may be useful as a first screen to identify novel biomarkers. Yet, in order to apply the terms "prognostic" or "predictive", this needs to be applied to individuals by determining the sensitivity and specificity of the respective marker. ROC analyses with AUC determinations and other statistical tools need to be applied towards this end.
2. Beyond the overall conceptual critique spelled out above, the authors could strengthen their findings by utilizing bioinformatic tools and publicly available databases (TCGA, GEO, ICGA etc.) to further substantiate PV-1 as a prognostic marker (on the population level). For instance, they could analyze differential expression of PV-1 between metastatic and non-metastatic samples by extracting gene expression data or perform Kaplan-Meier survival analysis on PV-1 expression levels.
3. The authors should clarify whether PV-1 is being proposed as a prognostic marker (indicative of the disease outcome regardless of treatment) or a predictive marker (indicative of response to specific treatment), particularly, in the context of luminal breast cancer.
4. The sample sizes are relatively small, particularly for soft tissue sarcomas (STS) where only 5-10 samples per histotype were analyzed. This limited sample size may not yield sufficient statistical power to draw reliable conclusions. Stratification by histotype in STS and inconsistent criteria for selecting luminal breast cancer (luminal BC) patients could introduce bias. The seven histotypes of STS exhibit varying metastatic potential (DOI: 10.20517/2394-4722.2022.138). Additionally, the selection criteria of patients who relapsed within 2 years versus those who remained disease-free for 5 years may also introduce bias in luminal BC. These factors collectively reduce the statistical power of multivariate analysis, potentially leading to unreliable results.
5. The authors should clarify whether PV-1 is being proposed as a prognostic maker (indicative of the disease outcome regardless of treatment) or a predictive marker (indicative of response to specific treatment), particularly, in the context of luminal breast cancer.
6. The absence of CD31 staining on cancer tissue makes it difficult to evaluate the quality of the analysis of %PV-1+/CD31+

cells. Throughout the manuscript, only mono PV-1 staining is presented. To demonstrate that the frequency of PV-1+/CD31+ cells is associated with metastatic progression (as in Fig 1A, 1B and 1C), representative CD31 staining should also be included (Fig 1B, 2B and 3C).

7. The conclusion regarding the endothelial barrier dysfunction and increased permeability in luminal breast cancer, ccRCC, and STS tumor tissues is overstated. The increase in PV-1+cell frequency is not sufficient to conclude endothelial barrier dysfunction and increased permeability. It is essential to combine more experimental methods to demonstrate the endothelial barrier dysfunction or increased permeability.

8. There are no functional studies to demonstrate the biological role of PV-1 in metastasis, particularly in the context of luminal BC, ccRCC or STS. This weakens the claim that PV-1 may be a marker for metastatic progression and metastasis free survival.

9. It is imperative to account for other confounding factors, such as differences in initial treatment response, treatment adherence (in luminal BC), patient comorbidities, and subsequent treatment or life-style changes.

10. Luminal breast cancer is heterogenous. The authors should specify whether PV-1+ cells are uniformly predictive across both Luminal A and Luminal B subtypes.

11. The immunohistochemical staining images have relatively low resolution. Without proper labelling, it is difficult to distinguish PV-1+ area in the non-metastatic samples shown in Fig. 1B, 2B and 3C.

As a service to authors, EMBO provides authors with the possibility to transfer a manuscript that one journal cannot offer to publish to another EMBO publication. The full manuscript and if applicable, reviewers reports are automatically sent to the receiving journal to allow for fast handling and a prompt decision on your manuscript. For more details of this service, and to transfer your manuscript to another EMBO title please click on Link Not Available

Referee #1 (Remarks for Author):

This is a well-conducted and highly interesting study that analyzes the potential role of PV-1 as a marker for distant metastasis in various solid tumors. The study presents a clear hypothesis, utilizes appropriate laboratory and statistical methodologies, and is well-written and clearly presented. PV-1 is known as a marker of endothelial permeability, and the authors have previously demonstrated its involvement in the dissemination of colorectal cancer. In this study, they show that the frequency of PV-1-positive cells is a predictor of metastasis-free survival in luminal breast cancer, clear cell renal cell carcinoma (ccRCC), and certain histotypes of soft tissue sarcomas.

The manuscript offers compelling findings that, if validated in future studies, could significantly aid clinicians and enhance diagnostic and therapeutic strategies.

I have only a few minor comments:

1. What do the authors think about the possibility of assessing blood levels of PV-1? Although blood levels do not provide certainty about the protein's origin, it would be interesting to evaluate this in future prospective studies due to the simplicity of blood sampling. I would appreciate the authors' thoughts on this matter.

Thank you for this question. We have started analyzing the plasma of CRC patients in a prospective study to evaluate whether we can predict the development of distal metastases. We are comparing the presence of PV-1 with circulating tumor DNA. We have very interesting preliminary results, but the study is ongoing. Thus, we think PV-1 may become a circulating marker of metastases.

2. The results are of great interest for all three types of tumors, although, as mentioned by the authors in the discussion, they require validation. However, in my opinion, the results on soft tissue tumors are somewhat less robust due to the small sample size related to the subtype. I recommend reducing the emphasis on the findings related to this type of tumor.

We have put the results on soft tissue sarcomas as preliminary data in supplementary table and figures.

3. Line 21: The authors reported "cell renal cell carcinoma" instead of "clear cell renal cell carcinoma."

Thank you. We have corrected it.

Referee #2 (Comments on Novelty/Model System for Author):

Elevated levels of PV-1 in the primary tumor have been previously shown to double the risk of developing distant metastasis in colorectal cancer patients. In this study, the frequency of PV-1 in endothelial cells was identified as a potential prognostic marker for distant metastases and metastasis free survival in luminal breast cancer, clear cell renal cell carcinoma and in histotype stratified soft tissue sarcomas. The presented results are novel and indicate that analysis of PV-1 expression may provide prognostic information. Therefore, this study is of clinical interest. However, further validation is required to confirm these findings and the title of the manuscript should be modified accordingly.

The major shortcoming of this study is the relatively low number of samples analysed. Especially in histotype stratified soft tissue sarcomas, the only one sample was representing non-metastatic cancers in LMS and UPS groups and therefore, these results should be considered as very preliminary rather than as compelling evidence that PV-1 is a prognostic marker of distant metastases in these STS subtypes.

We agree with the reviewer and we have clearly stated that these are preliminary results and have moved them in a supplementary figure. However, we would like to emphasize that, given their rarity, it is unrealistic to compare a case series of STS with breast and kidney cancer. Moreover, the study analyzes over 50 STS cases, and if we combine MPNST, LMS, UPS, and SFT, we have more than 20 cases. Interestingly, MPNST, LMS, UPS, and SFT are, compared to the other histotypes in the study, those with a more pronounced tendency for exclusively hematogenous metastasis; the potential significance of PV-1 in these histotypes is entirely consistent with their biology and clinical behavior. Additionally, the fact that there is only 1 non-metastatic case of UPS and LMS reflects the typical epidemiology and clinical history of these histotypes, which frequently metastasize.

In addition, it is unclear why the number of axillary lymph nodes having cancer cells was not presented, as it is commonly used as a prognostic indicator in breast cancer. Therefore, the advantage of analyzing PV-1 biomarker expression instead of the number of lymph nodes with cancer cells is not clear. For example, can PV-1 expression levels in endothelial cells of primary tumors identify those 30 % of patients without lymph node involvement at diagnosis who have increased risk of disease relapse and distant metastases?

We used a Cox model including the main known prognostic factors – specifically, the number of positive lymph nodes and the tumor size –, but the results were not significant. Anyway, the HR of PV-1 is basically unchanged in this analysis - 1.49 (95%CI 0.99-2.22) – with a p-value of 0.05. We included this table in the paper as Supplementary Table 1. However, since there was a correlation between the tumor size and the number of positive lymph nodes (Pearson $r = 0.52$), we believed that it would be better to have only one variable recapping both the tumor size and the number of positive lymph nodes, and that is why we used the tumor Stage in the paper.

	HR (95%CI)	P value
PV-1	1.49 (0.99-2.22)	0.05
PgR	0.92 (0.79-1.08)	0.31
Ki67	1.30 (0.91-1.85)	0.15
Tumor size	0.95 (0.64-1.41)	0.79
PLNs	1.04 (0.95-1.14)	0.36

Suppl. Table 1. Hazard Ratios, 95% confidence intervals and p-values obtained in the Cox regression model for distant metastasis-free survival. The percentage of PV-1+ cells, Ki67, and PgR are expressed in deciles. The tumor size is expressed in cm. PLNs: number of positive lymph nodes.

Regarding the second point, unfortunately, only 10 patients were without nodal involvement. Nevertheless, on these we performed two Cox models (the 1st one including PV-1, PgR, Ki67 and tumor stage; the 2nd one including PV-1, PgR, Ki67 and tumor size) :

Patients without nodal involvement (n=10) - Cox Model n. 1

	HR (95%CI)	P value
PV-1	3.97 (0.57-27.57)	0.16
PgR	0.99 (0.95-1.02)	0.48
Ki67	1.08 (0.99-1.17)	0.10
Tumor stage II	0.85 (0.02-45.03)	0.93

Hazard Ratios, 95% confidence intervals and p-values obtained in the Cox regression model for distant metastasis-free survival in BC patients without nodal involvement (n=10). The percentage of PV-1+ cells is expressed in deciles; the tumor stage HR is referred to tumor stage II vs. tumor stage I as reference. It is worth noticing that no patient without nodal involvement had tumor stage III.

Patients without nodal involvement (n=10) - Cox Model n. 2

	HR (95%CI)	P value
PV-1	7.02 (0.74-66.98)	0.09
PgR	0.93 (0.81-1.06)	0.27
Ki67	1.29 (0.87-1.91)	0.20
Tumor size	2.20 (0.63-7.68)	0.22

Hazard Ratios, 95% confidence intervals and p-values obtained in the Cox regression model for distant metastasis-free survival in BC patients without nodal involvement (n=10). The percentage of PV-1+ cells is expressed in deciles; the tumor size is expressed in mm.

The tendency for the effect of PV-1 is confirmed in both models, but due to the low sample size, the p-value was not significant. However, by plotting the PV-1 values for relapsed patients (metastatic) (1) and non-metastatic (0) patients in this group, we obtained the following graph, which shows that metastatic patients have higher values of PV-1 than non-metastatic patients:

Also, it is not fully clear what advantage the analysis PV-1 expression provide over measurement of tumor size or tumor grade in ccRCC.

In many of the analyzed ccRCC patients the levels of PV-1 are high even if they are not metastatic. As ccRCC is a tumor that can have a recurrence even decades after primary tumor resection¹⁻⁵, it would be interesting to evaluate whether they will become metastatic at a longer follow up time. This would be highly predictive of their metastatic potential and identify those patients that should be followed up for periods longer than the canonical 5-10 years.

At present, there is a lot of detailed information about metastatic processes for example in luminal breast cancer and ccRCC and therefore, it is an oversimplification to conclude that mechanism underlying dissemination remain poorly understood (Introduction, page 3, lines 68 and 69 as well as 83-84). The authors should state more specifically which dissemination processes they are referring to and provide references to support the conclusion.

We agree with the reviewer. The introduction has now been revised accordingly.

Referee #3 (Remarks for Author):

The authors have previously shown that PV-1 may be a prognostic marker for distant metastasis in CRC. The present work is an extension of this earlier work in other solid tumors such as luminal BC, ccRC and in STS. The authors employed 134 human tumor tissue samples to examine the frequency of PV-1+ endothelial cells and its association with metastatic status. However, the manuscript heavily relies on limited FFPE staining and statistical analysis of %PV-1+/CD31+ cells and survival analysis, which raises significant concerns about the robustness of the findings. In order to further develop their work, the authors are encouraged to consider the following:

1. The most critical conceptual limitation of the study is that the authors did not apply proper standards of a prognostic/predictive study. Biomarkers are not to be validated by comparing cohorts. The terms "prognostic markers" and "predictive markers" apply to individuals and not to populations. Population analyses may be useful as a first screen to identify novel biomarkers. Yet, in order to apply the terms "prognostic" or "predictive", this needs to be applied to individuals by determining the sensitivity and specificity of the respective marker. ROC analyses with AUC determinations and other statistical tools need to be applied towards this end.

We agree that for using those terms a ROC analysis should be used. Here are the results for the prediction of metastases in luminal breast cancer patients at 5 years, based only on the PV-1 continuous value. The model is a classical Logistic Regression classifier and the classification report is produced together with the ROC curve and its AUC:

	precision	recall	f1-score	support
0.0	0.88	0.58	0.70	12
1.0	0.77	0.94	0.85	18
accuracy			0.80	30
macro avg	0.82	0.76	0.78	30
weighted avg	0.81	0.80	0.79	30

Although the AUC is quite good it should be noted that these results are obtained by testing the classifier on the entire dataset, which was also used to train the model. Therefore, there is the risk of overfitting. This cannot be avoided, due to the low sample size. For this reason, we did not include such analyses in the paper. However, if you consider it necessary, we could add these data to the supplementary materials, to support our claim about PV-1 being a prognostic factor. Here is the resulting ROC curve for the ccRCC, using a Logistic Regression classifier with only PV-1 as predictor:

The result is worse because in ccRCC also non-metastatic patients tend to have high values of PV-1. The impact of PV-1 on risk of metastases is more evident in the Cox model as this has an impact on the time needed to develop metastases (higher PV-1 means higher risk, hence shorter time to develop metastases).

2. Beyond the overall conceptual critique spelled out above, the authors could strengthen their findings by utilizing bioinformatic tools and publicly available databases (TCGA, GEO, ICGA etc.) to further substantiate PV-1 as a prognostic marker (on the population level). For instance, they could analyze differential expression of PV-1 between metastatic and non-metastatic samples by extracting gene expression data or perform Kaplan-Meier survival analysis on PV-1 expression levels.

Thank you for your suggestion. We performed Kaplan-Meier survival analysis on PV-1 expression levels using KM plotter tool⁶⁻⁸A and we included these results as Supplementary Figure 1.

- mRNA gene chip data: by selecting Luminal A or Luminal B BC patients (treated with endocrine therapy) we obtained results that are consistent with ours (DMSF: distant-metastasis free survival; RFS: relapse free survival) (Suppl. Fig. 1A)

Luminal A

DMSF n=313

PLVAP (221529_s_at)

RFS n=481

PLVAP (221529_s_at)

Luminal B

DMSF n=200

RFS n=321

- Protein expression of PLVAP in breast cancer: Tang et al 2018 dataset⁸ (n= 65 tumor samples). OS plot: the p-value is not significant, but the trend is consistent with our analysis. However, we were unable to select only Luminal BC patients, as the analysis also includes HER2 and TNBC subtypes. (Suppl. Fig. 1B)

- mRNA RNA-seq data: we selected Kidney renal clear cell carcinoma (n=117) and we calculated RFS: the p-value is not significant, but the trend is consistent with our analysis (Suppl. Fig. 1C).

3. The authors should clarify whether PV-1 is being proposed as a prognostic marker (indicative of the disease outcome regardless of treatment) or a predictive marker (indicative of response to specific treatment), particularly, in the context of luminal breast cancer.

Please see reply to comment #1. However, we corrected in the manuscript the term and only used prognostic as there was no specific reference to the treatment.

4. The sample sizes are relatively small, particularly for soft tissue sarcomas (STS) where only 5-10 samples per histotype were analyzed. This limited sample size may not yield sufficient statistical power to draw reliable conclusions. Stratification by histotype in STS and inconsistent criteria for selecting luminal breast cancer (luminal BC) patients could introduce bias. The seven histotypes of STS exhibit varying metastatic potential (DOI: 10.20517/2394-4722.2022.138). Additionally, the selection criteria of patients who relapsed within 2 years versus those who remained disease-free for 5 years may also introduce bias in luminal BC. These factors collectively reduce the statistical power of multivariate analysis, potentially leading to unreliable results.

Reviewer raised an important point, which is at the core of dealing with sarcoma complexity: rarity of these diseases and heterogeneity of the histotypes. Expanding the sarcoma cohort is necessary for drawing conclusions. For these reasons we have moved this data into supplementary material. However, we would like to emphasize that the study analyzes over 50 STS cases, and if we combine MPNST, LMS, UPS, and SFT, we have more than 20 cases. Interestingly, MPNST, LMS, UPS, and SFT are, compared to the other histotypes in the study, those with a more pronounced tendency for exclusively hematogenous metastasis; the potential significance of PV-1 in these histotypes is entirely consistent with their biology and clinical behavior.

Probably there is a misunderstanding in the demographic table for BC: the follow-up time was 5 years for everyone, but those who developed metastases relapsed in a shorter time (i.e. within two years), and this is what is reported in the table. There was not a selection based on the time of metastases. Different time to metastases was among the principal inclusion criteria for the luminal breast cancer cohort in order to identify additional prognostic factors in patient with primary endocrine resistance.

5. The authors should clarify whether PV-1 is being proposed as a prognostic maker (indicative of the disease outcome regardless of treatment) or a predictive marker (indicative of response to specific treatment), particularly, in the context of luminal breast cancer.

Please see reply to comment #3.

6. The absence of CD31 staining on cancer tissue makes it difficult to evaluate the quality of the analysis of %PV-1+/CD31+ cells. Throughout the manuscript, only mono PV-1 staining is presented. To demonstrate that the frequency of PV-1+/CD31+ cells is associated with metastatic progression (as in Fig 1A, 1B and 1C), representative CD31 staining should also be included (Fig 1B, 2B and 3C).

We have added the slides including also CD31 (see new figures).

7. The conclusion regarding the endothelial barrier dysfunction and increased permeability in luminal breast cancer, ccRCC, and STS tumor tissues is overstated. The increase in PV-1+cell frequency is not sufficient to conclude endothelial barrier dysfunction and increased permeability. It is essential to combine more experimental methods to demonstrate the endothelial barrier dysfunction or increased permeability.

We use PV-1 as a proxy of increased permeability based on several previous studies in animal models which were paralleled by human observations. For instance, in animal studies after oral administration of *Salmonella*, PV-1 was upregulated and *Salmonella* was found in the liver. By using a genetic mouse model in which we impede PV-1 upregulation (through the use of β -catenin gain of function only in endothelial cells) we show that in this model *Salmonella* is unable to reach the liver any longer⁹. Similarly, upon high fat diet feeding we showed that PV-1 increased expression correlated with increased permeability as demonstrated by translocation of FITC-dextran administered orally into the peripheral circulation, or by fluorescent dye release in the gut after its intravenous injection as evidence by CELLVIZIO endoscopy^{10,11}. Higher PV-1 was observed also in human studies on NASH patients¹⁰. Our studies in a CRC mouse model similarly showed that PV-1 was associated with increased permeability, bacterial and tumor translocation into the liver¹².

8. There are no functional studies to demonstrate the biological role of PV-1 in metastasis, particularly in the context of luminal BC, ccRCC or STS. This weakens the claim that PV-1 may be a marker for metastatic progression and metastasis free survival.

We think this could be the subject of another study.

9. It is imperative to account for other confounding factors, such as differences in initial treatment response, treatment adherence (in luminal BC), patient comorbidities, and subsequent treatment or life-style changes.

The endocrine treatment in luminal BC patients is administered as adjuvant therapy so the response evaluable is relapse vs no relapse and this is the principal selection criteria for the 2 groups. Subsequent treatment after relapse would not be of interest for the objective of this study. Unfortunately, we do not have detailed information on life- style changes, but we have data regarding comorbidities. Of the 30 luminal BC patients, 16 had no comorbidities. Eleven reported only one comorbidity (dyslipidemia n=4; arterial hypertension n=3; bronchial asthma n=1; vascular disease n=1; hyperthyroidism n=2), while three patients had more than one comorbidity (dyslipidemia, arterial hypertension, type II diabetes mellitus, vascular disease: n=1; dyslipidemia, arterial hypertension, ischemic heart disease: n=1; arterial hypertension, hypothyroidism: n=1). After adding the presence or absence of any comorbidities as a variable to the Cox model, the results remained consistent, and the presence or absence of comorbidities was not significant. We included these data as Supplementary Table 2.

	HR (95%CI)	P value
PV-1	1.47 (1.03-2.10)	0.03
PgR	0.90 (0.77-1.05)	0.18
Ki67	1.03 (0.99-1.08)	0.11
Comorbidities	1.61 (0.44-5.95)	0.47
Tumor stage II	0.73 (0.14-3.81)	0.71
Tumor stage III	1.40 (0.25-7.98)	0.70

Suppl. Table 2. Hazard Ratios, 95% confidence intervals and p-values obtained in a Cox regression model for distant metastasis-free survival including the percentage of PV-1+ cells, the percentage of PgR and Ki67 expression, the tumor stage and the presence of any comorbidities. The percentage of PV-1+ cells, and that of PgR and Ki67 expression are expressed in deciles. The reference for tumor stages II or III is tumor stage I.

In conclusion, these comorbidities have no impact on the risk of metastasis.

Regarding ccRCC as well, it was not possible to retrieve detailed information about lifestyle changes, and there was no data in the literature correlating comorbidity factors with the onset of metastases. The only one that could play a role was lipid metabolism (dyslipidemia). Of the 52 ccRCC patients, 23 had no comorbidities, while 10 reported a single comorbidity (dyslipidemia n=3; arterial hypertension n=5; Obstructive Sleep Apnea Syndrome (OSAS) n=1; type II diabetes mellitus n=1), and 19 patients had multiple comorbidities. Adding the presence or absence of any comorbidity as a variable to the Cox model did not alter the results, and the presence/absence of comorbidities was not significant. We included these data as Supplementary Table 3.

	HR (95%CI)	P value
PV-1	1.29 (1.03-1.62)	0.02
Tumor size	1.45 (1.18-1.79)	<0.005
Tumor grade	4.17 (1.59-10.94)	<0.005
Tumor stage	0.36 (0.08-1.59)	0.18
Comorbidities	1.75 (0.66-4.66)	0.26

Suppl. Table 3. Hazard Ratios (HR), 95% confidence intervals and p-values obtained in a Cox regression model for distant metastasis-free survival in ccRCC patients, including the percentage of PV-1+ cells, the tumor size, the tumor grade, the tumor stage and the presence of any comorbidities. The percentage of PV-1+ cells is expressed in deciles. The tumor size is expressed in cm. The tumor grade HR is referred to grades III-IV vs. grades I-II as reference, and the tumor stage HR is referred to tumor stage pT3 vs. pT1-2 as reference.

In conclusion, these comorbidities have no impact on the risk of metastasis.

10. Luminal breast cancer is heterogenous. The authors should specify whether PV-1+ cells are uniformly predictive across both Luminal A and Luminal B subtypes.

Please consider this graph showing that in both luminal A and B, metastatic patients had higher percentage of PV-1+ cells (Mann-Whitney test: Luminal A: p=0.088, Luminal B: p=0.023):

Also, we performed two additional Cox models on the two subgroups of Luminal A and Luminal B separately, and obtained the same trend for PV-1 although we could not reach a significant p-value, probably due to the low sample size in both subtypes. This is why we considered them together. Additionally, the classification of subtypes was determined solely through immunohistochemistry (based on Ki67 percentage, tumor grading and progesterone receptor expression (PgR)) not according to the more accurate molecular definition.

Only Luminal A:

	HR (95%CI)	P value
PV-1	10.67 (0.46-246.17)	0.14
PgR	1.12 (0.71-1.77)	0.63
Ki67	0.11 (0.01-2.20)	0.15

Only Luminal B:

	HR (95%CI)	P value
PV-1	1.43 (0.95-2.15)	0.09
PgR	0.96 (0.77-1.18)	0.68
Ki67	1.35 (0.79-2.33)	0.27

Hazard Ratios, 95% confidence intervals and p-values obtained in the Cox regression model for distant metastasis-free survival. The percentage of PV-1+ cells, PgR and Ki67 are expressed in deciles.

11. The immunohistochemical staining images have relatively low resolution. Without proper labelling, it is difficult to distinguish PV-1+ area in the non-metastatic samples shown in Fig. 1B, 2B and 3C.

Thank you, we have increased the resolution.

REFERENCES

1. Abara E, Chivulescu I, Clerk N, Cano P, Goth A. Recurrent renal cell cancer: 10 years or more after nephrectomy. *Can Urol Assoc J*. Apr 2010;4(2):E45-9. doi:10.5489/cuaj.829
2. Lordan JT, Fawcett WJ, Karanjia ND. Solitary liver metastasis of chromophobe renal cell carcinoma 20 years after nephrectomy treated by hepatic resection. *Urology*. Jul 2008;72(1):230.e5-6. doi:10.1016/j.urology.2007.11.134
3. Nagai T, Igase M, Ochi M, et al. [Multiple metastases from renal carcinoma 15 years after nephrectomy]. *Nihon Ronen Igakkai Zasshi*. Nov 2007;44(6):747-51. doi:10.3143/geriatrics.44.747
4. Riviello C, Tanini I, Cipriani G, et al. Unusual gastric and pancreatic metastatic renal cell carcinoma presentation 10 years after surgery and immunotherapy: A case report and a review of literature. *World J Gastroenterol*. Aug 28 2006;12(32):5234-6. doi:10.3748/wjg.v12.i32.5234
5. Roser F, Rosahl SK, Samii M. Single cerebral metastasis 3 and 19 years after primary renal cell carcinoma: case report and review of the literature. *J Neurol Neurosurg Psychiatry*. Feb 2002;72(2):257-8. doi:10.1136/jnnp.72.2.257
6. Gyórfy B. Survival analysis across the entire transcriptome identifies biomarkers with the highest prognostic power in breast cancer. *Comput Struct Biotechnol J*. 2021;19:4101-4109. doi:10.1016/j.csbj.2021.07.014
7. Gyórfy B. Integrated analysis of public datasets for the discovery and validation of survival-associated genes in solid tumors. *Innovation (Camb)*. May 6 2024;5(3):100625. doi:10.1016/j.xinn.2024.100625
8. Tang W, Zhou M, Dorsey TH, et al. Integrated proteotranscriptomics of breast cancer reveals globally increased protein-mRNA concordance associated with subtypes and survival. *Genome Med*. Dec 3 2018;10(1):94. doi:10.1186/s13073-018-0602-x
9. Spadoni I, Zagato E, Bertocchi A, et al. A gut-vascular barrier controls the systemic dissemination of bacteria. *Science*. Nov 13 2015;350(6262):830-4. doi:10.1126/science.aad0135
10. Mouries J, Brescia P, Silvestri A, et al. Microbiota-driven gut vascular barrier disruption is a prerequisite for non-alcoholic steatohepatitis development. *J Hepatol*. Dec 2019;71(6):1216-1228. doi:10.1016/j.jhep.2019.08.005
11. Sorribas M, Jakob MO, Yilmaz B, et al. FXR modulates the gut-vascular barrier by regulating the entry sites for bacterial translocation in experimental cirrhosis. *J Hepatol*. Dec 2019;71(6):1126-1140. doi:10.1016/j.jhep.2019.06.017
12. Bertocchi A, Carloni S, Ravenda PS, et al. Gut vascular barrier impairment leads to intestinal bacteria dissemination and colorectal cancer metastasis to liver. *Cancer Cell*. May 10 2021;39(5):708-724.e11. doi:10.1016/j.ccell.2021.03.004

7th Mar 2025

Dear Maria,

Thank you for the submission of your revised manuscript to EMBO Molecular Medicine.

We have now received the reports from the 2 referees who reviewed your revised manuscript. Referee #1 already reviewed your initial submission (as referee #1), whereas referee #2 is a new referee, who was asked to focus on your answers to the initial referees' concerns.

As you will see below, while this referee acknowledges that several initial concerns have been addressed, he/she also regrets that some important points have not been satisfactorily addressed.

We have discussed this report within the team, and we would like to invite further revisions of the manuscript to address all referees' points, with the exception of the last, as also previously discussed (experimental evidence to support the claim that PV-1 contributes to the haematogenous spread of tumour cells through endothelial barrier dysfunction). This could instead be addressed by adequate discussion and toning down the claim.

As EMBO Press usually encourages one single round of revisions, please be aware that this will be the last chance for you to address the referees' concerns. The revised manuscript will once again be subjected to review, and we cannot guarantee a positive outcome at this stage.

Moreover, please address the following editorial requests:

1. Please provide a complete author checklist, which you can download from our author guidelines (<https://www.embopress.org/page/journal/17574684/authorguide#submissionofrevisions>). Please insert information in the checklist that is also reflected in the manuscript. The completed author checklist will also be part of the RPF.
2. Reagents and Tools Table: Please download and fill our Reagents and Tools Table template (.docx), which you can find in our author guidelines: <https://www.embopress.org/page/journal/14693178/authorguide#structuredmethods>. When submitting your revised manuscript, please do not include the Reagents and Tools Table in the Methods section of the manuscript but upload it as a separate file choosing the file type "Reagent Table".
3. Suppl. Fig 1-4 should be renamed Figure EV1 - EV4 and also uploaded as separate, high resolution figure files. The legends should be added to the manuscript text, after Table 2, following the main figure legends and under the heading "Expanded View Figure Legends"
4. The suppl. tables should be renamed Table EV1 - EV4 and uploaded as as separate files.
5. Please address the queries of our copy editors in the figure legends:
 - Please note that the exact p values are not provided in the legend of figure 2C
 - Please note that information related to n is missing in the legends of figures 1C, 2C, supplementary figure 4C
 - Please note that the measure of center for the error bars needs to be defined in the legends of figures 1C, 2C, supplementary figure 4C
6. Please provide source data for the main figures. Our source data coordinator will contact you to discuss which figure panels we would need source data for and will also provide you with helpful tips on how to upload and organize the files.
7. Please provide up to 5 keywords.
8. Author contributions: CRediT has replaced the traditional author contributions section because it offers a systematic machine readable author contributions format that allows for more effective research assessment. Please remove the Authors Contributions from the manuscript and use the free text boxes beneath each contributing author's name in our system to add specific details on the author's contribution. More information is available in our guide to authors.
9. Every published paper now includes a 'Synopsis' to further enhance discoverability. Synopses are displayed on the journal webpage and are freely accessible to all readers. They include a short stand first (maximum of 300 characters, including space) as well as 2-5 one-sentences bullet points that summarizes the paper. Please write the bullet points to summarize the key NEW findings. They should be designed to be complementary to the abstract - i.e. not repeat the same text. We encourage inclusion of key acronyms and quantitative information (maximum of 30 words / bullet point). Please use the passive voice. Please attach

these in a separate file or send them by email, we will incorporate them accordingly.

Please also suggest a visual abstract to illustrate your article as a PNG file 550 px wide x 300-600 px high. A cropped portion of this image will serve as thumbnail for the table of content on our webpage.

10. Please provide The Paper Explained: EMBO Molecular Medicine articles are accompanied by a summary of the articles to emphasize the major findings in the paper and their medical implications for the non-specialist reader. Please provide a draft summary of your article highlighting

As part of the EMBO Publications transparent editorial process initiative (see our Editorial at <http://embomolmed.embopress.org/content/2/9/329>), EMBO Molecular Medicine will publish online a Review Process File (RPF) to accompany accepted manuscripts.

In the event of acceptance, this file will be published in conjunction with your paper and will include the anonymous referee reports, your point-by-point response and all pertinent correspondence relating to the manuscript. Let us know whether you agree with the publication of the RPF and as here, if you want to remove or not any figures from it prior to publication. Please note that the Authors checklist will be published at the end of the RPF.

I look forward to receiving your revised manuscript.

With kind regards,

Lise

**** Reviewer's comments ****

Referee #1 (Remarks for Author):

The authors' answers are comprehensive and the additional analyses have improved the manuscript and resolved the doubts raised in the first review.

Referee #4 (Comments on Novelty/Model System for Author):

1. Statistical tests for Cox regression and ROC analysis are not adequate.
4. No pre-clinical model system is described in this study. Only patient samples.

Referee #4 (Remarks for Author):

Reviewer #2

The authors have addressed several concerns raised by Reviewer #2. However, the most critical part regarding the Cox regression is not adequately addressed as the statistics are not properly performed and are not unbiased. In addition, it is unclear whether the results presented in the tables are from univariate or multivariate Cox regression. Ideally, univariate Cox regression should be performed for all clinical variables available for each tumour type. A forward stepwise multivariate Cox regression (and a multivariate Cox regression with backward stepwise elimination) is then performed with all significant covariates from the univariate Cox test. The covariates included (or retained) in the final model can then be considered as independent prognostic factors. If two variables are correlated, the weakest is removed from the model in an unbiased manner in forward or backward Cox regression. Therefore, it is not appropriate to remove positive lymph nodes from the analysis just because "there was a correlation between tumour size and number of positive lymph nodes". Furthermore, the P value of PV-1 in Suppl. Table 1 does not appear to be significant ($P > 0.05$), as indicated by a 95% CI below 1.

Reviewer #3

The authors have addressed many of the concerns raised by Reviewer #3. However, there are still a number of critical issues that need to be addressed prior to publication:

- The receiver operating characteristic (ROC) curve and AUC determinations should be analysed as a time-dependent ROC, as this is a survival analysis. In addition, the ROC analyses for PV-1 should be compared with other known prognostic factors for each cancer type to assess whether or not PV-1 outperforms them. This information should be included as main and supplementary figures.
- Although the authors performed Kaplan-Meier survival analysis on PV-1 expression levels using the KM Plotter tool, the ROC analysis still needs to be validated in large cohorts of publicly available databases such as TCGA and GEO "to further substantiate PV-1 as a prognostic marker".
- All Kaplan-Meier survival analyses of PV-1 expression levels shown in Suppl. Fig. 1 show the same trend towards higher metastasis with higher PV-1 expression. However, the authors should justify why overall survival is much better for those luminal A breast cancer, clear cell renal cell carcinoma and sarcoma patients with tumours with high PLVAP expression, given that the sample size for overall survival is much larger than for relapse-free survival (RFS). These conflicting data should be discussed in the Discussion section to highlight the limitations of PV-1 expression as a prognostic factor.
- This study speculates that PV-1 contributes to the haematogenous spread of tumour cells through endothelial barrier dysfunction and increased permeability in luminal breast cancer, ccRCC and STS tumour tissues. However, experimental evidence to support such claims for each of these tumour types is still lacking.

***** Reviewer's comments *****

Referee #1 (Remarks for Author):

The authors' answers are comprehensive and the additional analyses have improved the manuscript and resolved the doubts raised in the first review.

Referee #4 (Comments on Novelty/Model System for Author):

1. Statistical tests for Cox regression and ROC analysis are not adequate.
4. No pre-clinical model system is described in this study. Only patient samples.

Referee #4 (Remarks for Author):

Reviewer #2

The authors have addressed several concerns raised by Reviewer #2. However, the most critical part regarding the Cox regression is not adequately addressed as the statistics are not properly performed and are not unbiased. In addition, it is unclear whether the results presented in the tables are from univariate or multivariate Cox regression. Ideally, univariate Cox regression should be performed for all clinical variables available for each tumour type. A forward stepwise multivariate Cox regression (and a multivariate Cox regression with backward stepwise elimination) is then performed with all significant covariates from the univariate Cox test. The covariates included (or retained) in the final model can then be considered as independent prognostic factors. If two variables are correlated, the weakest is removed from the model in an unbiased manner in forward or backward Cox regression. Therefore, it is not appropriate to remove positive lymph nodes from the analysis just because "there was a correlation between tumour size and number of positive lymph nodes". Furthermore, the P value of PV-1 in Suppl. Table 1 does not appear to be significant ($P > 0.05$), as indicated by a 95% CI below 1.

We thank the reviewer for the suggestion, which we have implemented with the help of an expert statistician (now included in the authors, Stefanos Bonovas). For both luminal breast cancer and ccRCC, we performed univariable Cox regression analyses for all clinical variables, followed by multivariable Cox regression with backward stepwise elimination to identify independent prognostic factors. Notably, variables with a p-value < 0.20 in the univariable Cox regression were considered eligible for inclusion in the backward elimination process. This relaxed threshold was chosen to ensure that potentially relevant predictors were not prematurely excluded. These new results confirm our previous findings and are presented in comprehensive summary tables with clear reference to the analysis: univariable analyses shown on the left and the corresponding multivariable model results on the right. These tables (Table 2 and Table 5) have replaced panel C of Figure 1 and Figure 2 respectively. Suppl. Tables 1-3 have been removed. These analyses have also been clarified in the Methods

section. We performed the same analyses also for STS and the results are presented in Appendix Tables S2 and S3. Thus, we have removed the old Supplementary Figure 2B-C and the panel C of Suppl. Figure 4 (now Figure EV5).

Reviewer #3

The authors have addressed many of the concerns raised by Reviewer #3. However, there are still a number of critical issues that need to be addressed prior to publication:

- The receiver operating characteristic (ROC) curve and AUC determinations should be analysed as a time-dependent ROC, as this is a survival analysis. In addition, the ROC analyses for PV-1 should be compared with other known prognostic factors for each cancer type to assess whether or not PV-1 outperforms them. This information should be included as main and supplementary figures.

The reviewer correctly pointed out that a time-dependent ROC analysis is more appropriate for time-to-event (survival) data. Accordingly, we performed time-dependent ROC analyses at five years for each clinical variable. We chose the 5-year time point because it represents a clinically relevant threshold, commonly used in oncology to assess long-term outcomes and recurrence risk. The corresponding AUC values are reported in Table 3 and Table 6 for luminal breast cancer and ccRCC, respectively. The ROC curves are shown in Figure EV1 and Figure EV4. We performed time-dependent ROC analysis also for clinical variables of STS patients belonging to LMS, MPNST, SFT, and UPS histotypes. ROC curve is shown in Figure EV5C.

- Although the authors performed Kaplan-Meier survival analysis on PV-1 expression levels using the KM Plotter tool, the ROC analysis still needs to be validated in large cohorts of publicly available databases such as TCGA and GEO "to further substantiate PV-1 as a prognostic marker".

We thank the reviewer for this comment. We agree that validating ROC analysis in large, publicly available datasets would strengthen the prognostic value of PV-1. However, we would like to clarify a few key points and the rationale behind the approach we used.

To compare gene expression data with the percentage of PV-1+/CD31+ cells observed in our patient samples, it is crucial to use patient datasets with treatment regimens and molecular classifications comparable to ours, and to perform a time-dependent ROC analysis at five years to ensure consistency with our clinical context.

We apologize if this was not made sufficiently clear in the original version of the manuscript, but the KM Plotter tool we used derives its data from multiple publicly available GEO datasets, as now specified in the revised Materials and Methods section. This tool performs a meta-analysis by combining datasets that include information on tumor subtype, treatment, RFS and DMFS, thereby allowing us to analyze them as a unified dataset. In Tables 1 and 2 shown below, we report all datasets included in the analysis along with the number of samples (patients) per dataset used to generate DMFS and RFS Kaplan-Meier curves.

Table 1. Datasets and number of patients per dataset included in the DMFS Kaplan-Meier analysis.

mRNA gene chip breast cancer			
DMSF			
Luminal A		Luminal B	
Dataset	n	Dataset	n
GSE17907 (n=54)	1		
GSE19615 (n=115)	19	GSE19615 (n=115)	2
GSE26971 (n=276)	157	GSE26971 (n=276)	82
GSE2990 (n=102)	15	GSE2990 (n=102)	17
GSE3494 (n=251)	30	GSE3494 (n=251)	21
GSE45255 (n=139)	16	GSE45255 (n=139)	21
GSE6532 (n=82)	34	GSE6532 (n=82)	29
GSE9195 (n=77)	41	GSE9195 (n=77)	28
total	313	total	200

Table 2. Datasets and number of patients per dataset included in the RFS Kaplan-Meier analysis.

mRNA gene chip breast cancer			
RFS			
Luminal A		Luminal B	
Dataset	n	Dataset	n
GSE12093 (n=136)	78	GSE12093 (n=136)	52
GSE1456 (n=159)	21	GSE1456 (n=159)	40
GSE16391 (n=55)	17	GSE16391 (n=55)	12
GSE17705 (n=196)	118	GSE17705 (n=196)	55
GSE17907 (n=54)	1		
GSE19615 (n=115)	19	GSE19615 (n=115)	2
GSE21653 (n=240)	27	GSE21653 (n=240)	22
GSE26971 (n=276)	65	GSE26971 (n=276)	24
GSE2990 (n=102)	19	GSE2990 (n=102)	19
GSE3494 (n=251)	32	GSE3494 (n=251)	24
GSE45255 (n=139)	9	GSE45255 (n=139)	11
GSE6532 (n=82)	34	GSE6532 (n=82)	32
GSE9195 (n=77)	41	GSE9195 (n=77)	28
total	481	total	321

However, a limitation of KM Plotter is that it does not disclose which specific samples were included in the analysis. This is further complicated by the fact that many of these datasets have been reanalyzed, meaning that the original GEO datasets were reprocessed in new studies using different analytical pipelines, updated annotations, or integration with additional data. Consequently, when we attempted to reanalyze these datasets, we could not reproduce the KM Plotter results, as the exact samples used were not identifiable. This lack of transparency prevents us from performing an independent, time-dependent ROC analysis using those data.

We have also searched for other suitable datasets from TCGA and METABRIC, but unfortunately none of them contained all the variables required for our analysis. For instance, the fields RFS, DMFS, and essential clinical data such as treatment information are often missing, incomplete, or not standardized. Even when accessing the versions of these datasets

available through cBioPortal, it was not possible to retrieve all the information necessary for our analyses.

Given these constraints, we turned to ROC Plotter, an online bioinformatics tool that evaluates the prognostic or predictive potential of gene expression by constructing ROC curves using clinical outcome data—such as relapse-free survival (RFS)—from large annotated datasets. Similar to KM Plotter, ROC Plotter performs meta-analysis by combining major publicly available datasets with annotated tumor subtypes, treatment, and RFS data, treating them as a unified cohort (Fekete J & Gyorffy B, *Int J Cancer*, 2019 Dec 1;145(11):3140-3151). The tool uses five-year relapse-free survival (RFS) as a clinical endpoint to construct ROC curves, classifying patients based on their outcome: those without relapse within five years and those with relapse within five years. Gene expression is assessed as a continuous predictor of clinical outcome. Importantly - and this is why we chose to use this tool - all datasets included in KMplotter were used for this analysis (except for GSE21653).

Using ROC Plotter, we focused on Luminal A (n=637) and Luminal B (n= 253) patients treated with endocrine therapy, to best matched our own study cohort. The resulting AUC values were: AUC = 0.533 (luminal A) and AUC = 0.608 (luminal B). These results have now been added as Figure EV3 in the revised manuscript.

To further assess the robustness of KM Plotter results, we analyzed a well-characterized and complete GEO dataset: GSE7390 (Desmedt et al., 2007), which includes both molecular subtype classification and DMFS data. This dataset consists exclusively of untreated patients, which limits its comparability with our study cohort, but still offers useful insights. It contains mRNA gene chip data from 78 luminal A patients (19 with distant metastases) and 45 luminal B patients (18 with distant metastases). Kaplan-Meier analysis, stratifying patients into high and low *PLVAP* expression groups using an automatic cut-off (see revised Material and Methods section), revealed significant differences in DMFS KM curves ($p = 0.033$ for luminal A and $p = 0.001$ for luminal B). We also performed a time-dependent ROC analysis at five years using GSE7390 dataset and obtained an AUC of 0.549 for luminal A and 0.715 for luminal B (see below, Figure 1).

To validate these findings, we repeated the DMFS Kaplan-Meier analysis using KM Plotter restricting the analysis to the GSE7390 dataset. The results were comparable: $p = 0.02$ for luminal A and $p = 0.013$ for luminal B. The DMFS KM curves were nearly identical to ours, although not exactly the same, as KM Plotter included a slightly different number of patients: 89 luminal A and 41 luminal B (see below, Figure 2A). These results reinforce the reliability of the KM Plotter tool, which is particularly important given its widespread use in meta-analyses of survival data.

Notably, even in this untreated cohort, high *PLVAP* expression was associated with shorter DMFS, highlighting the potential of *PLVAP* as a prognostic marker and suggesting that patients with elevated expression levels may be more prone to earlier metastatic progression.

To extend this finding, we performed KM Plotter analysis on untreated patients from multiple GEO datasets (GSE11121, GSE19615, GSE2990, GSE3494, GSE45255, GSE69031, and GSE7390). Again, we observed a significant association between high *PLVAP* expression and

shorter DMFS in luminal B patients (n=120, p = 0.03), and a trend toward significance in luminal A patients (n=276, p = 0.22) (see below, Figure 2B).

We opted not to include these data in the main manuscript, as the cohort is made up of untreated patients, while our PV-1 IHC data refer to patients treated with endocrine therapy. For ccRCC, the situation is even more challenging. Although a TCGA dataset exists, we encountered multiple inconsistencies—such as DMFS dates following dates of death—which made any reliable analysis impossible.

In conclusion despite our thorough efforts, publicly available datasets present important limitations in terms of clinical annotation, treatment information, and endpoint standardization. Additionally, mRNA expression levels do not necessarily correlate with protein expression levels, making direct comparisons between our data and publicly available datasets particularly challenging. Therefore, we believe any interpretation of similarities or discrepancies between our results and publicly available data should be made with caution.

Figure 1. Time-dependent ROC analysis for luminal breast cancer at five years and Kaplan-Meier survival analysis of *PLVAP* expression levels using the GEO dataset GSE7390. A, Time dependent ROC curve (left) and distant-metastasis free survival (DMSF) KM curve (right) analyzed using mRNA gene chip data from luminal A patients (n=78). B, Time dependent ROC curve (left) and DMFS KM curve (right) analyzed using mRNA gene chip data from luminal B patients (n=45). Patients were stratified into two groups based on *PLVAP* expression.

Figure 2. Kaplan-Meier survival analysis of *PLVAP* expression levels in untreated luminal breast cancer patients using KM plotter. A, Distant-metastasis free survival (DMSF) analysis based on *PLVAP* expression in the GSE7390 dataset, including luminal A patients (n=89) (left) and luminal B patients (n=41) (right). B, DMSF analysis based on *PLVAP* expression across multiple GEO datasets (GSE11121, GSE19615, GSE2990, GSE3494, GSE45255, GSE69031, and GSE7390), including luminal A patients (n=276) (left) and luminal B patients (n=120) (right).

- All Kaplan-Meier survival analyses of PV-1 expression levels shown in Suppl. Fig. 1 show the same trend towards higher metastasis with higher PV-1 expression. However, the authors should justify why overall survival is much better for those luminal A breast cancer, clear cell renal cell carcinoma and sarcoma patients with tumours with high PLVAP expression, given that the sample size for overall survival is much larger than for relapse-free survival (RFS). These conflicting data should be discussed in the Discussion section to highlight the limitations of PV-1 expression as a prognostic factor.

We thank the reviewer for this insightful comment. We acknowledge that, in some publicly available datasets (e.g., KM Plotter), higher PLVAP mRNA expression appears to correlate with better overall survival (OS) in luminal A breast cancer, clear cell renal cell carcinoma (ccRCC), and soft tissue sarcoma. However, our study specifically focuses on distant metastasis-free survival (DMFS) as a clinically meaningful endpoint, supported by our own protein-level data from well-annotated clinical cohorts. It is important to note that OS is influenced by many variables beyond metastatic spread, including comorbidities, subsequent lines of therapy, and variations in clinical management—all of which may confound the prognostic significance of PLVAP expression. In contrast, DMFS more directly reflects the biological process in which we propose PV-1 may be involved. Moreover, mRNA expression do not necessarily reflect protein expression, due to post-transcriptional regulation and tissue-specific expression patterns. As noted in our previous response to reviewer, we believe that any comparison between our findings and results from public datasets should be interpreted with caution.

- This study speculates that PV-1 contributes to the haematogenous spread of tumour cells through endothelial barrier dysfunction and increased permeability in luminal breast cancer, ccRCC and STS tumour tissues. However, experimental evidence to support such claims for each of these tumour types is still lacking.

We addressed this point in the discussion. We considered PV-1 as a marker of increased permeability based on several publications from our and other groups now reported in the discussion. We have however stated in the limitations of the study that we acknowledge that we have not functionally demonstrated the role of PV-1 in barrier dysfunction in the analyzed tumors.

2nd Jul 2025

Dear Maria,

Thank you for submitting your revised study. Referee #4 reviewed your revised manuscript and is satisfied with the revisions. I will therefore be able to accept your manuscript once the following editorial concerns are addressed:

1/ Manuscript text:

- Please remove the underlined text and only keep in track changes mode any new modification.
- Please provide the 5 keywords in the manuscript.
- Disclosure and competing interests statement: Please add "Maria Rescigno is an EMM editorial board member, etc"

2/ Figures:

- Please correct the nomenclature in the legends of Expanded View Figures to Figure EV1, etc. instead of Expanded View Figure 1, etc.
- Please indicate n= in the figure legend of Fig. EV1, EV4, EV5B, C.
- "Supplementary material" should be replaced by "Appendix" in the text.

3/ Source Data: please group them into folder per figure, where each folder would have a separate subfolder - one per each panel.

4/ Checklist:

The subsection "Studies involving specimen and field samples" was filled in, but I don't think it applies to your study. Please check and correct if needed.

5/ Thank you for providing a synopsis text. Please also suggest a visual abstract to illustrate your article as a PNG file 550 px wide x 300-600 px high. A cropped portion of this image will serve as thumbnail for the table of content on our webpage.

6/ As part of the EMBO Publications transparent editorial process initiative (see our Editorial at <http://embomolmed.embopress.org/content/2/9/329>), EMBO Molecular Medicine will publish online a Review Process File (RPF) to accompany accepted manuscripts. We note that you agree with the publication of the RPF.

I look forward to receiving your revised manuscript.

With kind regards,

Lise

***** Reviewer's comments *****

Referee #4 (Remarks for Author):

The authors have addressed the primary concerns raised by this reviewer. Consequently, the manuscript can be accepted for publication.

The authors addressed the remaining editorial issues.

9th Jul 2025

Dear Maria,

I am pleased to inform you that your manuscript is accepted for publication and is now being sent to our publisher to be included in the next available issue of EMBO Molecular Medicine.

If you have any questions, please do not hesitate to contact the Editorial Office.
Thank you for your contribution to EMBO Molecular Medicine!

With kind regards,

Lise
